# Measurement Report: Aircraft Observations of Ozone, Nitrogen Oxides, and Volatile Organic Compounds over Hebei Province, China

Sarah E. Benish[1], Hao He[1], Xinrong Ren[1,2], Sandra J. Roberts[3], Ross J. Salawitch[1,3], Zhanqing Li[1,4], Fei Wang[4,5], Yuying Wang[6], Fang Zhang[4], Min Shao[7], Sihua Lu[7], Russell R. Dickerson[1]

[1] Department of Atmospheric and Oceanic Science, University of Maryland, College Park, MD 20742, USA.
[2] Air Resources Laboratory, National Oceanic and Atmospheric Administration, College Park, MD 20740, USA.
[3] Department of Chemistry and Biochemistry, University of Maryland, College Park, MD 20742, USA.
[4] State Key Laboratory of Earth Surface Processes and Resource Ecology, College of Global Change and Earth System Science, Beijing Normal University, Beijing, 100875, China.
[5] Key Laboratory for Cloud Physics, Chinese Academy of Meteorological Sciences, Beijing, 100081, China.
[6] Key Laboratory for Aerosol-Cloud-Precipitation of China Meteorological Administration, School of Atmospheric Physics, Nanjing University of Information Science and Technology, Nanjing, 21004, China.
[7] College of Environmental Science and Engineering, Peking University, Beijing, 100871, China.

*Correspondence to*: Sarah E. Benish (sebenish@umd.edu)

**Abstract.** To provide insight into the planetary boundary layer (PBL) production of ozone ($O_3$) over the North China Plain, the Air chemistry Research in Asia (ARIAs) campaign conducted aircraft measurements of air pollutants over Hebei Province, China between May and June 2016. We evaluate vertical profiles of trace gas species including $O_3$, nitrogen oxides ($NO_x$), carbon monoxide (CO), and volatile organic compounds (VOCs) and relate to rates of $O_3$ production. This analysis shows measured $O_3$ levels ranged from 45 to 146 ppbv, with the peak median concentration (~92 ppbv) occurring between 1000 and 1500 m. The $NO_x$ concentrations exhibited strong spatial and altitudinal variations, with a maximum of 53 ppbv. Ratios of $CO/CO_2$ indicate the prevalence of low efficiency combustion from biomass burning and residential coal burning, but indicate some success of regional pollution controls compared to earlier studies in China. Concentrations of total measured VOCs reveals alkanes dominate the total measured volume mixing ratio of VOCs (68%) and sources include vehicular emissions, fuel and solvent evaporation, and biomass burning. Alkanes and alkenes/alkynes are responsible for 74% of the total VOC reactivity assessed by calculating the OH loss rates, while aromatics contribute the most to the total Ozone Formation Potential (OFP) (43%) with toluene, m/p-xylene, ethylene, propylene, and i-pentane playing significant roles in the aloft production of $O_3$ in this region. In the PBL below 500 m, box model calculations constrained by measured precursors indicate the peak rate of mean $O_3$ production was ~7 ppbv/hour. Pollution frequently extended above the PBL into the lower free troposphere around 3000 m, where $NO_2$ mixing ratios (~400 pptv) led to net production rates of $O_3$ up to ~3 ppbv/hour; this pollution can travel substantial distances downwind. The $O_3$ sensitivity regime is determined to be $NO_x$-limited throughout the PBL, while more VOC-limited at low altitudes near urban centers, demonstrating that control of both VOCs and $NO_x$ is needed to reduce aloft $O_3$ pollution over Hebei.

## 1 Introduction

Explosive urbanization and rapid industrialization contributed to high ground-level ozone ($O_3$) and particulate matter (PM) over the past several decades in the North China Plain (NCP) (Johnson et al., 2006; Ran et al., 2011; Shao et al., 2009; Zhang et al., 2014). Household burning of coal used for cooking and heating, emissions from gasoline, diesel, and liquified petroleum gas (LPG) vehicles, as well as large-scale burning of winter wheat residues in the NCP are some of the many sources responsible for $O_3$ precursors, such as nitrogen oxides ($NO_x=NO+NO_2$) and volatile organic compounds (VOCs) (Chen et al.,

2017; Long et al., 2016; Stavrakou et al., 2016). Ozone is harmful to both the human respiratory system (Bell et al., 2006;
Jerrett et al., 2009) and to photosynthetic processes by vegetation (Avnery et al., 2011; Reich and Amundson, 1985), while
some VOCs, such as benzene and chloroform, are known to be hemotoxic and carcinogenic (Environmental Protection Agency
- Integrated Risk Information System, 2003; Lan et al., 2004). Several studies using the NASA Ozone Monitoring Instrument
(OMI) have found reductions of some pollutants like sulfur dioxide ($SO_2$) over the NCP (He et al., 2012; Krotkov et al., 2016;
Li et al., 2010, 2017), but $NO_2$ pollution still remains severe in China (Figure 1a).

Ozone is created through the oxidation of NO by hydroperoxyl radicals ($HO_2$) and organic peroxy radicals ($RO_2$), products of
carbon monoxide (CO) and VOC oxidation. When one of these precursors is the limiting reactant, the rate of $O_3$ production is
considered VOC- or $NO_x$-sensitive (Finlayson-Pitts and Pitts, 1999; Sillman et al., 1990). The role of VOCs on the formation
of $O_3$ depends on the characteristics of the environment, including the main emission sources of primary pollutants and ambient
temperature (Pusede et al., 2014), and the interaction of aerosols within the PBL to reduce photolysis (An et al., 2019). High
aerosol concentrations have been shown to decrease photolysis and hinder summer surface $O_3$ formation by 25 ppbv on average
in Xi'an, China (Feng et al., 2016), which pose a challenge for pollution control strategies.

Natural emissions are the largest source of VOCs globally and react more efficiently with OH than most anthropogenic
compounds (Di Carlo et al., 2004), but exhibit a strong seasonal, diurnal, and spatial dependence (Li et al., 2013). Biogenic
VOCs have been found to play a significant role in the formation of $O_3$ at the surface (Ma et al., 2019; Zong et al., 2018) and
throughout the boundary layer in the NCP (Wang et al., 2008a), as well as influence production of $PM_{2.5}$ (Guo et al., 2014)
and secondary organic aerosols (SOA) (Wu et al., 2020b). In particular, isoprene has been estimated to account for 27% of the
total $O_3$ production in June 2010 in Beijing (Mo et al., 2018), suggesting the need to consider biogenic isoprene emissions in
formulating $O_3$ control strategies. Quantifying the abundance of $NO_x$ and the suite of VOC chemicals throughout the lower
troposphere is urgently needed to better understand the photochemistry of $O_3$ production in the NCP, which in turn will lead
to the development of successful mitigation strategies.

In-situ airborne measurements provide valuable information regarding the horizontal and vertical distributions of air pollutants
over a large spatial area. Airborne measurements are necessary to characterize air pollution over large cities, as well as
surrounding areas. Ozone and PM are produced throughout the planetary boundary layer (PBL), so aircraft observations can
lead to a more complete picture of pollution formation and transport than is available only from surface observations. While
several airborne campaigns have deployed to investigate the regionally transported pollution problem in East Asia, including
the NASA Korea-United States Air Quality Study (KORUS-AQ) (Al-Saadi et al., (2015),    see https://www-
air.larc.nasa.gov/missions/korus-aq/docs/White_paper_NASA_KORUS-AQ.pdf), that occurred at the same time as our
measurements, few airborne studies characterize the source region of severe smog within the Hebei Province region of China.

Through Chinese/American partnerships with Peking University, Beijing Normal University, and the University of Maryland,
we conducted a field campaign in Hebei Province, China called Air chemistry Research In Asia (ARIAs). The ARIAs

campaign was designed to characterize and quantify the composition of trace gases and aerosol optical properties over Hebei to improve tools used to evaluate the effectiveness of air pollution reduction policies. Since air pollution transport from Asia typically peaks in early to mid-spring (Liu et al., 2003), we hoped to provide detailed altitude profiles over the Asian source region to enable Lagrangian experiments with KORUS-AQ, but only two sustained transport events occurred (Peterson et al., 2019). Despite the infrequent transboundary pollution events, ARIAs observations generated valuable characteristic pollution signatures that helped describe combustion efficiency and its impact downwind (Halliday et al., 2019) to correct model biases of CO in global chemistry-climate models (Gaubert et al., 2020), and to show that MOPITT bias increases at high CO concentrations (Tang et al., 2020). Furthermore, ARIAs measurements characterized aerosol optical properties in the planetary boundary layer and free troposphere during clean and polluted conditions (Wang et al., 2018a), as well as used in the validation of MAX-DOAS profiles of $NO_2$, $SO_2$, HONO, HCHO, CHOCHO, and aerosols (Wang et al., 2019b).

The Ministry of Environmental Protection of the People's Republic of China reported six of the top ten cities with the worst air quality in 2016 were located in Hebei (including the capital city of Shijiazhuang). The North China Plain is one of the most polluted regions in the world, but implementation of pollution reduction measures through the Five-Year Plans has allowed for decreasing trends of many pollutants. In particular, Zhang et al. (2020) found an increased number of days of clean/light haze and a decreased number of days with heavy haze, along with a significant decline of $SO_2$ concentrations. Similarly, using observations from MODIS and OMI, Si et al. (2019) found AOD and $SO_2$ to decrease from 2006 to 2015, while $NO_2$ rose by 4.79% in the NCP during this period. While surface $NO_2$ decreased 20% from May 2014 to December 2018 throughout China, there are still a large number of measurement stations with increasing trends of $NO_2$ due to changes in meteorological conditions and aerosol emissions (Fan et al., 2020), illustrating the need for more research characterizing air pollution in this region. In this study, we analyze concentrations of $O_3$, $NO_x$, $NO_y$, CO, and VOCs obtained during 11 research flights between May and June 2016. The VOC chemical reactivity and impact on $O_3$ production is assessed and we utilize an observation-constrained box model to evaluate photochemical properties of the production of $O_3$ that occurs throughout the lower free troposphere.

## 2 Materials and Methods

### 2.1 Air sampling and analysis

The ARIAs campaign included 11 research flights from May-June 2016 in Hebei Province (Fig. 1b). Flight days were chosen based on meteorological conditions associated with smog events, such as higher temperatures, little cloud cover, low relative humidity, weak winds, and shallow PBL height (Tang et al., 2012). The Y-12 aircraft was based at Luancheng Airport (114.59°E, 37.91°N, 58 m above sea level (ASL)), located in southeast Shijiazhuang (population around 10 million), a major economic centre in Hebei, including pharmaceutical and textile industries, machinery and chemical manufacturing, construction, and electronics production. Flight sampling occurred east of the Taihang Mountains and the Y-12 flew vertical

spirals from ~300 m to ~3500 m over Shijiazhuang as well as three other locations: Julu (115.02°E, 37.22°N, 30 m ASL), Quzhou (114.96°E, 36.76°N, 40 m ASL), and Xingtai (114.36°E, 37.18°N, 182 m ASL) (see Table S1 for a description of flight paths, weather conditions, and statistics of measured trace gases).

Various instruments aboard the Y-12 aircraft collected trace gas, aerosol, and meteorological data. The aircraft instrumentation (Table 1) included different gas and particle sample inlets on the top of the fuselage and pressure/temperature/humidity sensors (Cloud Water Inertial Probe (CWIP), Rain Dynamics) installed under one wing of the aircraft (Fig. S1). Flight position data were recorded using a portable global positioning system (GPS) and the CWIP. The aircraft was equipped with the following trace gas analyzers: (1) a Picarro cavity ring down spectrometer (CRDS) for measurements of $CH_4$, $CO_2$, CO, and $H_2O$; (2) a

Thermal Electron Corporation (TECO) Model 49C UV absorption $O_3$ analyzer; (3) a TECO Model 43C pulsed fluorescence $SO_2$ analyzer; (4) a Los Gatos Research Model RMT-200 CRDS $NO_2$ analyzer; and (5) a TECO Model 42C $NO-NO_y$ analyzer. Power constraints and a converter issue led to limited $NO_y$, $NO_x$, and CO measurements during the campaign, particularly in the lower free troposphere (LFT). We remove observations of $NO/NO_y$ over three spiral locations due to limited measurements. Negative values indicate readings around the detection limit, usually at high altitudes. The aircraft was also equipped with an

inlet to measure aerosols up to ~5.0 µm diameter and aerosol optical properties, including a nephelometer (TSI Model 3563) to measure aerosol scattering, a particle soot absorption photometer (PSAP) to measure aerosol absorption, and an aethalometer (Magee Model AE31) and a Single-Particle Soot Photometer (SP2, Droplet Measurement Technologies) to measure black carbon. Observed aerosol optical properties have been summarized by F. Wang et al. (2018); further details on aircraft instrumentation are given by Ren et al. (2018).

Twenty-six whole air samples (WAS) were collected directly into 3.2 L fused silica lined electropolished stainless steel canisters (Entech Instrument Inc., Simi Valley, CA) at a variety of pressure altitudes from 400 m to 3500 m between 1:30 and 9:00 UTC (9:30 and 17:00 local time). The sampling period for the WAS canisters was approximately 1-2 minutes during the spirals. Samples were analyzed for 54 VOCs and 16 halocarbons. Since the halogenated species have negligible effects on $O_3$ production, we exclude these species from the analysis presented here. We also exclude 2 WAS canisters from this analysis

due to evidence of contamination after collection (Text S1). Limited samples collected over one province in one season may not be able to represent $O_3$ chemistry for all of China, but the scarcity of airborne VOC measurements in this region makes these data valuable for characterizing the composition of air throughout and above the PBL, demonstrating how the production of $O_3$ aloft differs from that at the surface.

     The VOC analytical techniques used by the College of Environmental Sciences and Engineering at Peking University (PKU)

in Beijing have been summarized in the past (Mo et al., 2015; Wang et al., 2010a), and we briefly describe the method here. The WAS canisters were cleaned following a standard sampling procedure, pressured with nitrogen and vacuumed three times to 2.6 Pa. The hydrocarbons were quantified using a gas chromatograph equipped with a mass selective detector (GC-MSD, Hewlett Packard 5975/7890, USA) and a flame ionization detector (FID) coupled with a cryofocusing pre-concentration system (Entech Instrument 7100A, Simi Valley, CA). This system used a Dean Switch™ (Agilent Technologies, Santa Clara, CA,

USA) to introduce the effluent into a DB-624 column (60 m × 0.25 mm × 1.8 μm; J&W Scientific, Folsom, CA, USA) with an MSD to separate and analyze C4–C12 hydrocarbons and halocarbons. A PLOT (Al/KCl) column (30 m × 0.25 mm × 3.0 μm; J&W Scientific) with an FID was used to measure the C2–C4 hydrocarbons.

The Photochemical Assessment Monitoring Stations (PAMS) (55 NMHCs) and Toxic Organic-15 (TO-15) standard mixtures were used to calibrate the GC-MSD/FID system that measured the C2–C12 VOCs. Samples with known concentrations of four VOCs (bromochloromethane, 1,4-difluorobenzene, chlorobenzene-d5 and 1-bromo-3-fluorobenzene) were used as internal standards for each sample to calibrate the system. The GC-MSD/FID system was calibrated at five concentrations, ranging from 0.5 to 8 ppbv, for each of these four compounds before sample analysis. Correlation coefficients, ranging from 0.987 to 0.999 showed that the integral areas of peaks were proportional to the concentrations of the target compounds. A gas standard (diluting from 1 ppmv to 2 ppbv) was measured each day to check the stability of the system. Summary statistics of the VOCs along with the method detection limit (MDL) (ranging from 0.002 to 0.027 ppbv) are reported in Table S2. Total uncertainty for VOC measurements reflects instrument noise, plus uncertainty in calibration standards, contamination, and pressurization. Best estimate of the total uncertainty is ±20% with 95% confidence due to uncertainties associated with airborne sampling platforms. Intercomparison experiments of VOC measurements between PKU and other laboratories showed good agreement (Liu et al., 2008b).

The Atmosphere-Aerosol-Boundary Layer-Cloud ($A^2BC$) Interaction Joint Experiment campaign collected meteorological, aerosol, and trace gas information from a ground-based site in Xingtai (114.36°E, 37.18°N, 182 m ASL) from May to December 2016 (Wang et al., 2018b, 2019b). An intensive observation period in May and June 2016 was conducted to coincide with ARIAs. Data from $A^2BC$ instruments used in our analysis include: (1) a $NO_x$ analyzer with a molybdenum converter (Ecotech model 9841A); (2) an infrared absorption CO analyzer (Ecotech model 9830A); and (3) a UV absorption $O_3$ analyzer (Ecotech model 9810A). Results of $NO_2$, $SO_2$, HONO, HCHO, CHOCHO, and aerosols derived from the Differential Optical Absorption Spectrometer (DOAS) are summarized by Yang Wang et al., (2019). The $A^2BC$ site is located in northwest Xingtai, nestled in the east foothill of the Taihang Mountains. Agricultural crops surround the site, consisting heavily of winter wheat, that is harvested, with the stubble burned in June (Liu and Si, 2011). Xingtai is a city with approximately 7 million people and is surrounded by industry including coal mining and coal-burning power plants, cement and steel industries, chemical processing, iron-smelting, and glass manufacturing.

## 2.2 Box Model Simulations

A box model called Framework in 0-Dimensional Atmospheric Modelling (F0AMv3.1) (Wolfe et al., 2016) is used to evaluate oxidation processes to understand $O_3$ photochemical production both at the surface and aloft. The box model simulations cover the Y-12 flight tracks during seven flights and daytime hours at the $A^2BC$ supersite in Xingtai (where the Y-12 conducted spirals) using the Carbon Bond Mechanism, version 6, revision 2 (CB6r2). Both the Y-12 flights and surface simulations define a physical loss lifetime of 24 hours to mitigate build-up of long-lived oxidation products over multiple days of integration.

For the ARIAs flight data, the model is constrained by 1-minute average observed concentrations of VOCs, $NO_2$, CO, and $O_3$. Due to the limited number of grab canisters per flight, VOCs are constrained based on the altitude of the sampling relative to the height of the PBL, which is determined using potential temperature and water vapor vertical profiles for each flight. All WAS canister data collected below the top of the PBL during a flight are averaged. Data from all of the WAS canisters for the entire campaign collected above the research flight's PBL are averaged for that flight. Periodic missing Y-12 $NO_2$ data due to internal auto-zeroing are linearly interpolated since gaps were short (~2 minutes). The chemical system defined by each set of observations is integrated 5 days forward in time, in 1-hour time steps with diurnal variation of solar zenith angle (SZA), in order for calculated reactive intermediates to achieve diel steady state. Reaction rate constants are calculated using aircraft measurements of pressure, temperature, and relative humidity. The SZA is determined based on the time and location of the aircraft, and used to calculate photolysis rates as described below.

For the $A^2BC$ surface data, the model is constrained by 5-minute average concentrations of VOCs, $NO_2$, CO, and $O_3$ on days that a flight occurred. For May 17, surface data for $NO_2$ is filled with 1-hour average data collected for other days of the month, due to missing surface measurements on this day. The average concentrations from the WAS canisters below 500 m are used as ground concentrations since $A^2BC$ did not measure VOCs at the surface. Similar to the flight data, the chemical system for the surface observations is integrated for 3 days forward in time, in 1-hour time steps with time-varying SZA, to reach diel steady state. Reaction rate constants are calculated from ground measurements of pressure, temperature, and relative humidity. Time and ground elevation are used to calculate the SZA, which controls photolysis frequencies as described below.

Photolysis frequencies, not measured during ARIAs or at the $A^2BC$ supersite, evolve over the course of a model step and are calculated by combining cross sections and quantum yields with solar spectra derived from the NCAR Tropospheric Ultraviolet and Visible (TUV) version 5.2 radiation model. At the start of the model run, input solar zenith angle, altitude or elevation, $O_3$ column, and surface albedo are used for linear interpolation across TUV lookup tables (F0AM's "hybrid" method). We use SZA and altitude/elevation from ARIAs/$A^2BC$ measurements and constant values for ozone column (325 DU) and surface albedo (0.17), which we estimate based on concurrent data from the OMI level-3 OMDOAO3e data product (https://disc.gsfc.nasa.gov/datasets/OMDOAO3e_003/summary?keywords=OMDOAO3e_003). A correction factor of 0.8, determined by trial and error, is used to scale j-values to better agree with the observed $NO/NO_2$ ratio.

The impact of aerosols on $O_3$ production depends on the optical properties as well as the vertical distribution (Dickerson et al., 1997; Kelley et al., 1995). In the presence of scattering and absorbing aerosols, photolysis frequencies will be altered, thus changing the $O_3$ formation and atmospheric oxidizing capability (Wu et al., 2020a). Previous research over China has shown that as AOD increases, the extinction effect of aerosols on photolysis frequencies decreases due to a higher proportion of scattering aerosols under high AOD conditions (Wang et al., 2019a). Optical depth, single scattering albedo, and angstrom exponent during ARIAs (see Wang et al., 2018a) are used in the TUV online calculator (https://cprm.acom.ucar.edu/Models/TUV/Interactive_TUV/) to assess the impact of aerosols on photolysis frequencies. Most of the aerosol particles during ARIAs were concentrated in the lowest 2 km of the atmosphere with a single scattering albedo

at 550 nm of 0.85 and an average AOD ~0.2. The impact of aerosol optical properties measured during ARIAs on photolysis frequencies is small compared to the default setting, so no additional adjustments are made to the model values.

The method described here to constrain VOCs introduces large uncertainty due to the sparsity of measurements obtained over a large area that potentially consists of a wide variety of chemical compositions. However, the production of $O_3$ aloft is not well characterized over Hebei, so our observations may help improve the understanding of air pollution for this region, despite

this limitation. Additionally, unlike a 3-dimensional chemical transport model, the box model simulations do not include advection or emissions. These processes, while important, are not included in the box model since $O_3$ precursors were measured and used to constrain the box model calculations. Box modelling is used to gain an understanding of $O_3$ production and its sensitivity to ambient levels of $NO_x$ and VOCs based upon measured meteorological parameters and the concentration of a wide variety of chemical species.

**2.2.1 Ozone Production and Sensitivity Calculations**

The photochemical production of $O_3$ during the daytime is determined by the production rate of $NO_2$ molecules from the $HO_2+NO$ and $RO_2+NO$ reactions minus the loss mechanisms (Finlayson-Pitts and Pitts, 1999). Thus, the net $O_3$ production rate, net($PO_3$) can be estimated following Equation 1:

$$net(PO_3) = k_{HO_2+NO}[HO_2][NO] + \sum_{i=1}^{n} k_{RO_{2i}+NO}[RO_{2i}][NO] - P(RONO_2)$$

$$-k_{OH+NO_2+M}[OH][NO_2][M] - k_{HO_2+O_3}[HO_2][O_3] - k_{OH+O_3}[OH][O_3]$$
$$-k_{O(^1D)+H_2O}[O(^1D)][H_2O] - L(O_3 + alkenes) \tag{1}$$

where $k$ denotes the different reaction rate constants and $RO_{2i}$ is the concentration of individual organic perxoy radicals. The terms subtracted from the production of $O_3$ are the loss mechanisms: the formation of nitrates, $P(RONO_2)$, the reaction of OH and $NO_2$ to form nitric acid, the reactions of OH and $HO_2$ with $O_3$, the reaction of $O(^1D)$ with $H_2O$, and the reactions of $O_3$ with alkenes. Additional terms not included here are the rate of $O_3$ loss by dry deposition and direct loss on aerosol surfaces

(dilution is the only physical loss in the current F0AM setup).

We evaluate the sensitivity of $O_3$ production to $NO_x$ and VOCs using the ratio of $L_N/Q$, where $L_N$ is the radical loss through reactions with NO and Q is the primary radical production (Kleinman, 2005a). When $L_N/Q$ is much less than 0.5, the $O_3$ production regime is $NO_x$-limited; when $L_N/Q$ ratio is much higher than 0.5, the regime is VOC-limited. Different environments can have varying amounts of organic nitrates that impact the cut-off value of $L_N/Q$, so this value could vary

around 0.5 (Kleinman, 2005b).

# 3 Results and discussion

## 3.1 Observations of nitrogen oxides, carbon monoxide, and ozone

Our observations confirm heavy loadings of air pollution over Hebei. Vertical profiles show peak median concentrations of NO (1.6 ppbv), $NO_2$ (4.4 ppbv) and $NO_y$ (25.7 ppbv) below 500 m with large variability (Fig. 2). Median concentrations of NO and $NO_2$ drop off gradually with altitude, while median $NO_y$ remains close to ~15 ppbv throughout most of the profile. Between 500 and 1000 m, sufficient levels of $NO_x$ are observed (median=3.8 ppbv), indicating continued production of $O_3$ in the PBL. Above 3000 m, median concentrations of NO and $NO_2$ fall to 350 pptv and 106 pptv, respectively (not measured simultaneously), still sufficient to produce $O_3$ as air parcels travels downwind. Median mixing ratios of $O_3$ and CO remain high (~80 ppbv and ~120 ppbv, respectively) throughout the altitudes sampled by the Y-12.

Unlike previous airborne studies over Beijing from 1994-2005 that found increased $O_3$ concentrations below 1 km with constant levels (~52 ppbv) between 1 and 2 km (Ding et al., 2008), our $O_3$ concentrations peaked between 1000 and 1500 m (median = 91.6 ppbv). Low ratios of $NO_x/NO_y$ (<0.30) indicate significant $O_3$ production had already occurred, but the strong correlation (R=0.71, Fig. S2) between 1-minute $NO_z$ ($NO_y$-$NO_x$) and $O_x$ ($O_3$+$NO_2$), an empirical estimate of the $O_3$ production efficiency (OPE), below 1500 m demonstrates moderate production of $O_3$ continued during sampling. The OPE of ~3.5 during ARIAs is smaller than the average OPE value of ~8 obtained during 2013 DISCOVER-AQ flights in Houston (Mazzuca et al., 2016), likely due to the higher $NO_x$ concentrations observed in Hebei than Texas.

Maps of $O_3$ and $NO_2$ on the Y-12 flight tracks (Fig. 3) show the largest concentrations around the spiral locations as well as between the three most northern cities, Shijiazhuang, Julu, and Xingtai. Regions of elevated $NO_2$ do not always correspond with high $O_3$ concentrations. The flight with the maximum observed $NO_2$ mixing ratio (35.3 ppbv) during ARIAs occurred on May 17 around 8:30 am LST. The aircraft was flying a flat transect at 500 m from Shijiazhuang to Julu when a large peak of $NO_2$, $CO_2$ (500 ppmv), and NO (15 ppbv) was sampled. Concentrations of $O_3$ were low during the time of the peak (~60 ppbv), indicating NO-$O_3$ titration, but $O_3$ levels were quite high (>90 ppbv) throughout the remainder of the flight. The maximum $O_3$ concentration (142.5 ppbv) was measured on May 21 during descent into Luancheng Airport in Shijiazhuang. Observations of $NO_x$ were not available for this flight, but elevated CO concentrations (565 ppbv) were observed. High concentrations of $O_3$ were also observed away from the large megacities. For instance, an $O_3$ plume (~125 ppbv) was sampled on June 6 at 1500 m over a more suburban area between Shijiazhuang and Julu with $NO_2$ levels ~500 pptv.

Vertical profiles of trace gases over the four spiral locations (Fig. 4) generally show the highest concentrations over the two largest cities, Shijiazhuang and Xingtai. These two megacities exhibit the greatest variability, below 500 m altitude, of all trace gases discussed here below 500 m. At 3000 m, Xingtai demonstrates the most $NO_2$ (~800 pptv), while the other spiral locations show ~300 pptv. Median profiles of $NO_y$ below 500 m are highest over Julu (27.6 ppbv). Median vertical profiles of CO are relatively consistent (~300 ppbv) below 2000 m over the spiral locations, while Julu shows the highest median concentration between 2500 and 3000 m (209.1 ppbv). Measurements of CO above Xingtai indicate a large spread in observations at all

altitudes from the lowest 500 m ($10^{th}$ percentile= 258 ppbv, $90^{th}$ percentile=1049 ppbv) up to 2000 m ($10^{th}$ percentile=97.7 ppbv, $90^{th}$ percentile = 135 ppbv). This variability may be partially explained by the possible burning of wheat straw during early summer 2016. Strong correlations between ethane and acetylene, two biomass burning markers (see Section 3.2), further suggest wheat residue burning over Xingtai. Median vertical profiles of $O_3$ below 500 m were 10-25 ppbv higher in Shijiazhuang (median=96.2 ppbv) than the other spiral locations. Concentrations of $O_3$ are generally stable or slightly increasing in the lowest 2000 m, and median $O_3$ is 75-80 ppbv even as high as 2500-3000 m. Xingtai shows the smallest variability of aloft $O_3$ levels above 2000 m, likely due to the position of this city on the leeward side of the Taihang Mountains.

The vertical profiles of $O_3$ compared to concurrent surface measurements in Xingtai indicates the $A^2BC$ site usually observed larger average concentrations than observed aloft, but this difference was highly dependent upon time of day (Fig. S3). The early afternoon profiles on May 8 showed average surface concentrations only slightly higher than the Y-12 measurements at ~400 m, while the mid-afternoon profiles on May 21 showed ~25 ppbv higher surface $O_3$ concentrations than Y-12 observations. At low altitudes (~700 m), the late morning flight (around 11:00 LST) on May 28 observed levels of $O_3$ ranging from 72-80 ppbv, comparable to average surface concentrations of 78 ppbv at the same time. By contrast, the afternoon flight (approximately 17:00 LST) at the same altitude later that day observed ~25 ppbv lower levels of $O_3$ compared to the surface (average=121 ppbv). All profiles on June 11 showed 10-30 ppbv lower average surface concentrations than measured during the Y-12 spirals.

The overall measured concentrations (1-second data, standard deviation, minimum, and maximum values) of $NO_x$, CO, and $O_3$ in this study are compared with other airborne studies in China including KORUS-AQ flights when outflow was directly from China (Table 2). Comparable to our range of $NO_x$ levels from concentrations near the detection limit to 53.2 ppbv, autumn flights in the Yangtze River Delta in 2007 documented large variability in $NO_x$ concentrations, ranging from 3 to 40 ppbv (Geng et al., 2009), while April 2006 observations in northern China similarly find a mean concentration ~5 ppbv (Wang et al., 2008b). The minimum CO concentration during ARIAs (80.5 ppbv) was measured in the lower free troposphere, which is a much smaller minimum concentration than reported by earlier studies. The warm-sector PBL air ahead of a cold front in April 2007 in Shenyang Province in northeast China found ~300 ppbv CO between 1000 and 4000 m (Dickerson et al., 2007), generally larger than most ARIAs profiles (except for Julu). The maximum value of CO during ARIAs (over 6 ppmv) agrees better with the literature, although there are few reported aircraft measurements of CO in Northeast Asia. Average and maximum $O_3$ concentrations during ARIAs were much higher than in other studies, but comparable to KORUS-AQ measurements from May 24-29 when the flow of air was direct from China. Since the majority of past airborne studies occurred over the sea areas during other seasons, it is not surprising that an urbanized environment like Hebei experienced much larger amounts of $O_3$ than previously reported.

The ratios between combustion tracers can be used to understand the source and efficiency. During high-efficiency combustion in modern power plants, fuel carbon is converted to $CO_2$ with near unit efficiency, resulting in low $CO/CO_2$ (<0.10%), while low-efficiency combustion (cold or smoldering processes or low-technology combustion) yields larger ratios. The regression

of 1-second CO against $CO_2$ (Fig. 5a) shows high linear correlation (R=0.90) and high ratios of $CO/CO_2$ (3.1%) together with large amounts of $SO_2$. These measurements are illustrative of low-efficiency fossil fuel combustion, likely from residential coal burning as these observations were all collected at ~500 m, and are compared to other studies in Table 3. Our results indicating the prevalence of low-efficiency combustion agree with KORUS-AQ airborne data over the West Sea with 2.8% $CO/CO_2$ (Tang et al., 2018), as well as with December 2017 surface measurements at Jingdezhen station in central China of 2.6% when air mass transport was from northern China (Xia et al., 2020). Compared to earlier studies in rural and urban areas of Beijing in the mid-2000s (Han et al., 2009; Wang et al., 2010b) and to 2011 measurements in Nanjing (Huang et al., 2015), the ARIAs $CO/CO_2$ ratio is 0.1-2.7% lower, evident of some success of regional pollution control strategies. By contrast, our $CO/CO_2$ ratio is higher than satellite-derived ratios over megacities that have implemented extensive pollution control measures (Silva et al., 2013). Similarly, compared to airborne measurements from the 2015 Wintertime INvestigation of Transport, Emissions, and Reactivity (WINTER) campaign in the Baltimore/Washington, D.C. region (Ren et al., 2018), our $CO/CO_2$ ratio is about a factor of 6 larger. Higher $CO/CO_2$ ratios (~6%) with less than 0.1 ppm $SO_2$, as seen briefly during three ARIAs flights, are more in line with emissions from burning of wheat straw in Hebei of ~6% (Cao et al., 2008), and other inefficient, biofuel combustion.

The $\Delta CO/\Delta NO_y$ ratio (equivalent to the slope in a CO vs. $NO_y$ plot) (Fig. 5b) is an indicator for distinguishing plumes with efficient $O_3$ formation, Typical values of this ratio are ~40 in background air and between ~4-7 in fresh emissions plumes in Houston (Neuman et al., 2009). The $\Delta CO/\Delta NO_y$ ratio of 23.5 measured during ARIAs indicates some photochemical aging and contributions from fossil fuel or biomass burning, but high values of CO, $NO_y$, and $SO_2$ suggests sampling of air parcels heavily influenced by power plants. The $CO/NO_x$ emission ratio (Fig. 5c) from ARIAs agrees with higher emission ratios of gasoline vehicles, while higher amounts of CO, $NO_x$, and $SO_2$ indicate coal burning from the residential sector or inefficient electric generating units. While most of these observations are reflective of the prevalence of low efficiency fossil fuel combustion, the aircraft sampled a plume on June 6 while flying spirals over Julu containing 0.9% $CO/CO_2$ and 0.4% $SO_2/CO_2$ (Fig. S4), likely due to a coal-burning power plant operating at high combustion efficiency, either using a sulfur scrubber or burning low sulfur fuel.

**3.2 Observations and sources of VOCs**

The total measured VOC mixing ratios ranged from 4 to 23 ppbv, largely dependent upon the altitude of collection, and was mostly dominated by alkanes (Fig. 6). The samples associated with the largest concentrations of $O_3$ were all collected at altitudes ~500 m during a period with stagnant high pressure. Generally, the samples collected below 500 m showed larger amounts of alkenes/alkynes and aromatics than canisters collected elsewhere in the PBL. The top VOCs ranked by mean volume mixing ratio (Table 4) shows that alkanes dominate the total measured VOC mixing ratio during ARIAs (68%), followed by alkenes/alkynes (17%), and aromatics (15%). The top 10 VOC species are C2-C5 alkanes, C2-C3 alkenes/alkynes,

benzene, and toluene. The observed mixing ratios of ethane and propane are 2.65 ppbv and 1.39 ppbv, respectively, which together accounts for ~52% of the total alkane mixing ratio.

The levels of ambient VOCs during ARIAs are generally lower than prior surface observations since measurements were taken in the PBL away from primary sources. Prior ground-based studies have similarly found alkanes to contribute the majority (>50%) of the total VOC concentration in late spring in the Beijing-Tianjin-Hebei region (Li et al., 2015; Tang et al., 2009; Yuan et al., 2013). The most abundant species during ARIAs are comparable to previous studies finding ethane, propane, and acetylene among the most prevalent, but likely have different sources based on the study location (Jia et al., 2016; Li et al., 2015; Mo et al., 2015; Tang et al., 2009). In the Beijing-Tianjin-Hebei region, ambient acetylene, ethylene, and other light alkanes have been attributed to emissions from gasoline vehicles (Li et al., 2015), while in Guangzhou, the widespread use of LPG has resulted in high levels of propane (Guo et al., 2011). Additionally, our observations have higher amounts of branched alkanes, such as 2,2,4-trimethylpentane and 2-methylheptane (both components of gasoline), but lower amounts of isoprene due to collection over mostly urban regions with lower ambient temperatures than the summer months. Since isoprene with a lifetime of hours (Seinfeld and Pandis, 2006) in the summer typically exhibits a strong vertical gradient in the PBL (Huang et al., 2017), we find the mean amount of isoprene measured during ARIAs is about 7 times lower than average May 2014 surface measurements in Beijing (Li et al., 2015), as well as ~200 pptv lower than June-July 2007 airborne measurements in the PBL in NE China (Xue et al., 2011). Next, we examine the potential sources contributing to observations of VOCs by comparing with ratios and correlations from known sources.

Since CO can be a marker for anthropogenically emitted hydrocarbons, particularly combustion products, we first use the ratios of various VOCs to CO to reveal insight into changes in emissions in the region. Ratios of VOCs to CO can vary substantially among cities (Baker et al., 2008; Warneke et al., 2007), but in general can provide details about fuel types and combustion efficiency between metropolitan regions. Despite ARIAs measurements sampling in close proximity to local VOCs sources, most VOCs do not correlate strongly with CO, reflective of the lack of common source signatures and some photochemical aging of the sampled airmasses. We report slopes of VOCs/CO in Table S2 when R>0.50. Ethane has the strongest correlation with CO (R=0.72) and the slope (2.5 pptv/ppbv) agrees well with ratios from urban areas of the United States in 1999-2005 (2.4 pptv/ppbv) (Baker et al., 2008) as well as with charcoal burning emission ratios (Andreae and Merlet, 2001). The ARIAs emission ratio of benzene/CO (1.8 pptv/ppbv) is slightly higher than found in urban regions of the United States (0.7, Baker et al., 2008) and Mexico City (0.93-1.20, Apel et al., 2010), likely due to higher emissions by widespread combustion of coal and agricultural residues (Zhang et al., 2015). By contrast, the ARIAs emission ratios of ethylene and acetylene to CO (2.9. and 1.4 pptv/ppbv, respectively) are lower than observed in urban areas in the United States (4.1 and 3.4 pptv/ppbv, respectively) and Mexico City (7.90-8.40 and 8.20-9.60 pptv/ppbv, respectively), where the dominant source was reported to be transportation-related (Baker et al., 2008). The lower ratio of ethylene/CO is comparable to emission ratios reported from charcoal burning (2.3 pptv/ppbv) (Andreae and Merlet, 2001).

Ethane is the most abundant VOC in this study and correlates well with indicators for biomass and coal burning (R>0.81), such as acetylene, ethylene, benzene, and $SO_2$. The ratio of acetylene to ethane (Fig. 7a) during ARIAs is 0.59, comparable to the ratio found in a plume of fresh biomass burning in Canada (Blake et al., 1994) and within the range of crop residue burning (~0.2-0.6) found in other studies in China (Chen et al., 2017). High ratios of benzene/propane (1.12) are comparable to dry grass combustion samples collected in the central Pearl River Delta (PRD) (1.6) (Wang et al., 2005) and further confirm the presence of VOCs due to biomass burning.

The C3 and C4 alkanes, including propane and the butanes, are the three main components of LPG and their correlation acts as an indicator for LPG leakage. In this study, a moderate correlation (R~0.50) is found between n-butane and propane and i-butane with n-butane. The ratio of n-butane/propane during ARIAs is 0.60, which agrees well with ratios from vehicle emissions (Liu et al., 2008a), but is lower than slopes measured in the PRD (2.1) (Lai et al., 2009), where VOCs originated from LPG leakage. Additionally, propane correlates well with acetylene and ethylene (Figure 7a), two well-known vehicular emission tracers.

Since acetylene and propane have comparable photochemical lifetimes with respect to OH attack, the ratio can be used to assess the relative importance of fossil fuel combustion and LPG leakage (Goldan et al., 2000). LPG contains propane but not acetylene (acetylene/propane<1) while combustion of fossil fuels commonly produces small amounts of propane relative to acetylene (acetylene/propane>1) (Conner et al., 1995; Gilman et al., 2013; Russo et al., 2010; Watson et al., 2001). In this study, the acetylene/propane ratio (Fig. 7a) is greater than 1, indicating emissions from vehicles (Fraser et al., 1998). These results suggest vehicles are largely responsible for the C3 and C4 alkanes as well as the C2 alkenes/alkynes observations.

The C5 alkanes and some C6 alkanes like 2,3-dimethylbutane and 2-methylpentane are found in vehicular exhaust and in gasoline vapor (Tsai et al., 2006). The i-pentane to n-pentane ratio is commonly used to identify the contributions of natural gas, vehicular emissions, and fuel evaporation since these alkanes have similar boiling points, vapor pressures, and reaction rate coefficients with OH. In areas heavily dominated by natural gas drilling, ratios lie between 0.82-0.89 (Gilman et al., 2013), while higher ratios are associated with vehicle emissions (2.2-3.8) and fuel evaporation (1.8-4.6) (Jobson et al., 2004; McGaughey et al., 2004; Russo et al., 2010; Wang et al., 2013). In this study, i-pentane and n-pentane are highly correlated (R=0.93), indicating a common source of these compounds. The slope is 10.3, higher than reported in previous studies in China (Li et al., 2019), and the large i-pentane concentrations are likely reflective of gasoline evaporation due to the extremely volatile nature of i-pentane. The influence of fuel evaporative emissions is further identified by strong correlations between C4-C7 alkanes and alkenes typical of fuel evaporative emissions. Strong correlations of many long-chain alkanes (C6-C7 and octane) with i-pentane (R>0.73 except for cyclohexane) but absence of correlations with acetylene indicates solvent evaporation may be another source of long-chain alkanes.

Typically, the ratio of cis-2-butene/trans-2-butene is used to determine the source of C4 alkenes (Li et al., 2015; Velasco et al., 2007). However, in this study, all measurements of cis-2-butene and trans-2-butene are below the detection limit, so assessing

the ratio and correlation is not possible. Previous studies in this region in China have attributed C4 alkenes to vehicular emissions (Li et al., 2015).

The correlation between the C7-C8 aromatics is strong (R>0.76) and revealing of typical signatures from incomplete combustion. The toluene/ethylbenzene ratio (10.7) is higher than traffic and urban emission ratios (~5-8), but closer to ratios associated with biomass burning (9.41) (Monod et al., 2001; Parrish et al., 1998). Toluene also correlates with all C7-C9 alkanes (R>0.64) and with i-pentane (R=0.85), compounds from diesel and gasoline evaporation. High levels of toluene reported in Hong Kong by Ho et al., (2004) were suggested to be emitted from gasoline evaporation, while Chan et al., (2006b) attributed the high toluene levels in different PRD cities to industrial solvent usage.

There is an excellent correlation (R>0.99) between o-xylene and m/p-xylene (Fig. 7b) and the slope (0.33) is comparable to the emission ratio found in a tunnel study (0.35) (Liu et al., 2008a). The o-xylene/ethylbenzene (0.60, Fig. 7b) slope is lower than vehicle exhaust emission ratios (1.2-1.8) (Conner et al., 1995; Jobson et al., 2004; Kirchstetter et al., 1996; Rogak et al., 1998; Sagebiel et al., 1996), but the correlation is extremely strong, suggesting the preferential loss of xylenes during transport due to their higher reactivity. These correlations and ratios suggest incomplete combustion from vehicular emissions and biomass burning are an important source of C7 and C8 aromatics.

The ratio between benzene/toluene (B/T) is a useful indicator to distinguish between vehicular emissions and other combustion sources. A ratio ~0.5 is often attributed to vehicular sources (Brocco et al., 1997; Perry and Gee, 1995), while ratios larger than 1 have been reported for coal or charcoal burning (Andreae and Merlet, 2001; Moreira Dos Santos et al., 2004). Benzene was observed at high mean ratios over Hebei (0.51 ppbv) and the average B/T ratio is 1.8±1.6 ppbv/ppbv. The correlation of some hydrocarbons can highlight the differences between B/T>1 (N=17) and B/T<1 (N=9). The correlation found between benzene and acetylene when all samples are grouped together (Fig. 7a) substantially improves just considering "traffic-related" samples (B/T<1) (R=0.93), suggesting a contribution of vehicular sources to benzene and acetylene measurements.

### 3.3 The effect of VOCs on ozone formation

In order to effectively reduce $O_3$ concentrations, it is crucial to understand the relative importance of individual VOCs in terms of the production of $O_3$ because each VOC exhibits different chemical reactivities. In this section, we present results using the loss rate of each VOC species with OH and ozone formation potential (OFP) assuming no influence of aerosols. Since the aerosol effect on $O_3$ formation is dependent upon time of day (solar zenith angle), meteorology, levels of local and neighboring aerosols, and the VOC/NOx ratio, the calculations presented here are simplified compared to the more complicated chemical composition of the atmosphere, but are still useful to help inform control strategies.

### 3.3.1 OH loss rate of VOC species

The calculation of the first-order loss rate of OH with different VOCs, termed OH reactivity, provides a measure of the potential to produce $HO_2$ and $RO_2$, key intermediate species in the production of $O_3$ (Stroud et al., 2008). Since the reaction with OH accounts for the majority of loss of most VOCs, the rate constant (obtained from the Master Chemical Mechanism version 3.3.1 (MCM3.3.1) and the National Institute of Standards and Technology (NIST) Chemical Kinetics database (www.kinetics.nist.gov/)) for the reaction between OH and various hydrocarbons reflects the overall reactivity of that

hydrocarbon (Finlayson-Pitts and Pitts, 1999). OH reactivity for each VOC species ($VOC_i$) is defined by Equation 2:

$$OHR(VOC_i) = k_{OH+VOC_i} * [VOC_i] \tag{2}$$

Where $k_{OH+VOC_i}$ is the reaction rate constant between OH and $VOC_i$. Among the VOC groups, alkanes and alkenes/alkynes both contribute the most to the total VOC reactivity, accounting for 37% each. Aromatics accounted for 26% of the total VOC reactivity. The relative contribution of the top 10 VOCs ranked by mean OH reactivity (Table 5) shows ethylene, propylene,

and isoprene among the top measured alkene species, together contributing ~33% to total OH reactivity. Among the alkanes, 2-methylpentane and i-pentane contribute the most (13%) to total OH reactivity, followed by the branched pentanes and propane. Aromatic compounds such as toluene and m/p-xylene constitute 13% to total OH reactivity. Previous ground-based summer studies in China have found larger contributions of isoprene to OH reactivity, ranging from ~10-30% (Li et al., 2015; Xue et al., 2017), than ARIAs (7.2%).

### 3.3.2 Ozone formation potential of NMHCs

Since OH reactivity only provides a qualitative identification of the most reactive species and does not reflect products and their production of further free radicals, we next consider the contribution to the formation of $O_3$ using ozone formation potential (OFP). The OFP of a VOC relies on the quantity maximum incremental reactivity (MIR), which represents the amount of $O_3$ formed from the addition of a small amount of the VOC species in interest under high $NO_x$ conditions. Values of MIR

(unit: g $O_3$ formed/ g VOC) have been calculated based on model simulations evaluated with smog chamber measurements (Carter, 2010, 1994). The OFP is calculated according to Equation 3:

$$OFP (VOC_i) = MIR_{VOC_i} * [VOC_i] \tag{3}$$

This method gives an estimate of only the first 24 hours after initial release. The median measured VOC/$NO_x$ ratio for all WAS canisters was 4.9 ppbv/ppbv. In comparison, the ratio of reactive organic gas to $NO_x$ (ROG/$NO_x$) in Los Angeles is 7.6

ppbv/ppbv (Carter, 1994). VOCs experience photochemical loss from emission sources near the surface to measured aloft concentrations. Estimation of OFP from aircraft observations throughout the PBL indicates how formation of $O_3$ may be different from previous surface studies.

To identify the major contributors to $O_3$ formation in this region, the 10 species with the highest mean OFP are listed in Table 4. Aromatic compounds are the largest contributor to total OFP (43%), followed by alkanes (30%) and alkenes/alkynes (27%). Toluene and ethylene make the largest contributions (19.6% and 15.7%, respectively) to total OFP. The high MIR of these compounds (MIR=4.0 g $O_3$/g VOC and 9.00 g $O_3$/g VOC, respectively) and large mixing ratios (4.9% and 5.7% of the total measured VOC volume mixing ratio) drives their important contribution to $O_3$ formation. The relatively short lifetime of ethylene (~1.4 days) combined with the large range of measured mixing ratios (0.18 to 3.54 ppbv) suggests sampling of air masses with little to moderate photochemical processing, indicating the large range of influence on OFP. The most reactive compound in terms of OFP is trans-2-butene (MIR=15.16 g $O_3$/g VOC), but its low concentration results in only 0.2% to total OFP. At the other extreme, ethane accounts for a relatively high percentage of total measured VOC volume mixing ratio (17.0%) yet only contributes 2.1% to OFP due to its low reactivity (MIR=0.49 g $O_3$/g VOC).

Previous studies in China report aromatics and alkenes account for the most OFP (Cai et al., 2010; Cheng et al., 2010; Jia et al., 2016; Liang et al., 2017; Wang et al., 2010a, 2016; Xie et al., 2008; Zheng et al., 2009). At a surface site in Beijing (May 2014), Li et al. (2015) found m/p-xylene, ethylene, toluene, propylene, and o-xylene are most influential to OFP, while at a ground station in Tianjin (August 2018), Han et al. (2020) found that ethylene, isoprene, toluene, m/p-xylene, and propylene were important contributors to OFP. Our study supports a larger contribution of anthropogenic VOCs than biogenic VOCs in spring, although summer studies indicate a major role for isoprene to the formation of $O_3$ in the NCP (Han et al., 2020; Zong et al., 2018). Since isoprene is mostly emitted by biogenic sources during the warmer summer months with strong solar radiation and when soil moisture is sufficient for plant growth, we expect isoprene to have a larger impact on $O_3$ production in the summer than during spring, the time of our study. J. H. Tang et al. (2007) concluded ethylene, toluene, and m/p-xylene are the main contributors to OFP during spring 2005 at the surface in the PRD, citing emissions from industry and vehicular exhaust. Our study agrees with past research in urban areas in China identifying the most reactive VOCs in terms of OFP; $O_3$ appears to be formed more slowly above the surface and in nonurban areas, but production is still substantial.

National measures for Chinese VOCs abatement were released in 2015, mainly focused on the reduction of anthropogenic VOCs from sources in the petrochemical industry, organic chemical industry, packaging printing, and industrial coating, not considering reactivity or chemical speciation (Li et al., 2018). A 2010 VOC emission inventory study concluded the top 15 OFP species (including m/p-xylene, toluene, propylene, o-xylene, and ethylbenzene) contributed 69% of total OFP, but only accounted for 30% of the total emission of VOCs by mass (Liang et al., 2017). Our analysis of the top 10 species ranked by mean OFP shows these compounds contribute 68% to total OFP but only represent 37% of the total volume mixing ratio. Li et al., (2018) classifies industrial coal burning, biomass burning, and motorcycles to the top three VOC emission sources in Shijiazhuang, but OFP is highest for furniture coating, automobile coating, diesel vehicles, fuel evaporation, and gasoline vehicles. These results confirm that reactivity scales and emissions rates should be considered together when formulating control strategies for $O_3$.

### 3.4 Photochemical ozone production rate and sensitivity

In this section, we describe calculated net photochemical production rates of $O_3$ using the box model constrained by aircraft observations. Ozone production rates calculated from the box model are high in major urban centres, particularly Shijiazhuang and Xingtai, but also between these cities (Fig. 8a). The highest rates (>10 ppbv/hour) are generally found closer to the surface, but in some instances upwards of 2000 m. The largest net production rate of $O_3$ (over 16 ppbv/hour) was located along the Taihang Mountains between Shijiazhuang and Xingtai. This large net production rate occurred ~2000 m on June 11, 2016 when NO, $NO_y$, $NO_2$, and $O_3$ were ~2 ppbv, ~18 ppbv, ~3 ppbv, and ~75 ppbv, respectively.

Vertical profiles of production, loss, and net rates of $O_3$ (Fig. 9) show that $HO_2$+NO made more $O_3$ than $RO_2$+NO during the campaign. The major loss of $O_3$ was due to the termination of $NO_2$ through its reaction with OH below 2500 m. Reaction with $O(^1D)$ is the main loss of $O_3$ above 2500 m. A maximum of net $O_3$ production for the mean profile was observed in the lowest 500 m of ~7 ppbv/hour. In the PBL between 1500-2000 m, where median NO and $NO_2$ were 534 and 625 pptv, respectively, $O_3$ production rates were ~4 ppbv/hour. In the lower FT from 2500 to 3000 m, peak net $O_3$ production rates still reached ~3ppbv/hour and were conducive to long-range transport.

Values of $L_N/Q$ (Fig. 8b) indicate production rates of $O_3$ are mostly $NO_x$-sensitive (i.e., $L_N/Q < 0.5$) in the PBL over Hebei and some of the largest net production rates of $O_3$ are associated with $NO_x$-sensitivity. In order to control aloft $O_3$ production that has the potential to be transported downwind, $NO_x$ is the most important precursor to control. However, at low altitudes near urban centres, the production rate of $O_3$ tends to be more VOC-sensitive (i.e., $L_N/Q > 0.5$), particularly during morning flights. In urban regions of China, an $O_3$ formation transition from VOC-limited at the surface to $NO_x$-limited at ~1 km has been documented (Chen et al., 2013; Han et al., 2020). Additionally, many studies conclude $O_3$ production in urban areas of China is VOC-sensitive in spring, while likely more $NO_x$-sensitive in more rural areas (Ran et al., 2011; Xue et al., 2013). Using updated emissions from a nonlinear joint analytical inversion of VOCs and $NO_x$ from the Ozone Mapping and Profile Suite Nadir Mapper (OMPS-NM) formaldehyde and OMI NO2 columns during KORUS-AQ in WRF-CMAQ, Souri et al. (2020) found the maximum daily 8 hour average surface $O_3$ over the NCP to increase by 4.56 ppbv, suggesting that emission control strategies on VOCs should be prioritized. Pusede et al., (2014) assessed the temperature dependence of emission control scenarios to lower $O_3$ in San Joaquin Valley, California and concluded reducing organic emissions at moderate and high temperatures with co-occurring $NO_x$ decreases will further diminish the number of $O_3$ violations. Thus, the control of $NO_x$ as well as VOCs may be necessary to control both aloft and near-ground $O_3$ production in the NCP.

### 4 Summary

High concentrations of $O_3$ and its precursors were pervasive over Hebei Province, China in Spring 2016. In this study, we quantify the composition and photochemical nature of the lower troposphere associated with smog events. Measurements of

trace gases including $O_3$, CO, $NO_x$, $NO_y$, and of aerosol optical properties were acquired in May and June 2016. Twenty-six samples analyzed for 54 VOCs were taken aboard a Y-12 research aircraft mostly in the PBL. Our observations confirm heavy loadings of pollution over Hebei.

The major conclusions of our study are:

1.    We observed high amounts $O_3$, ranging from 45 ppbv to 145 ppbv, with the highest values found over Shijiazhuang.
10        The highest $NO_x$ concentrations were observed over Xingtai below 500 m. The highest $NO_x$ and CO concentrations were 53.2 ppbv and 6054 ppbv, respectively. Ratios of $CO/CO_2$ indicate inefficient combustion from residential coal and biomass burning throughout the region but have decreased in China since the early 2000s suggesting the implementation of successful pollution control strategies.

2.    Concentrations of total measured VOCs reveals alkanes contribute the most by volume mixing ratio (68%), while
15        alkenes/alkynes and aromatics together supply the most (74%) to the calculated OH loss rate. Aromatics constitute most (43%) to the total calculated OFP and toluene, ethylene, m/p-xylene, propylene, and i-pentane play significant roles in the aloft formation of $O_3$ in this region. In contrast to other surface studies in summer, we find a lower contribution of biogenic sources (e.g. isoprene) to the formation of $O_3$ in the PBL. Sources of VOCs include vehicular emissions, biomass burning, and fuel and solvent evaporation.

3.    High amounts of $NO_x$ and VOCs throughout the PBL over nonurban parts of Hebei Province were found to generate $O_3$ at a peak mean rate of ~7 ppbv/hour below 500 m. The lower free troposphere (from ~2500 to ~3000 m) was also frequently polluted with CO and $NO_2$ averaging ~125 ppbv and ~140 pptv with peak net production rates of $O_3$ ~3 ppbv/hour, allowing for continued formation of $O_3$ as the air mass travels downwind. The $O_3$ production regime is found to be $NO_x$-limited throughout the PBL over Hebei, while more VOC-limited at low altitudes near urban centres.

Our measurements in spring 2016 over Hebei cannot represent all of China or the seasonal variation of $O_3$ photochemistry, but measurements from an airborne platform make a valuable addition to the understanding of one of the most polluted regions in China, and indeed the world. The photochemistry of $O_3$ production is highly dependent upon the interaction of radiation and aerosols within the PBL and future work is needed to assess optical properties of aerosols at wavelengths relevant to photolysis of $O_3$ to $O(^1D)$ and thus OH. We show that to improve air quality in Hebei Province, both $NO_x$ and VOCs from vehicles and
fuel evaporation should be targeted. While VOCs are already targeted for emission reduction in China, the egregious concentrations of $O_3$ observed in this study further confirm the formation of a reactivity-oriented control strategy is urgent.

**Author contributions**

The ARIAs campaign was supervised by RD, ZQ, and XR. XR, HH, and FW conducted the measurements on board the research aircraft and VOCs were analyzed by MS and SL. $A^2BC$ observations were collected by ZL, FW, YW, and FZ. SR
helped set up the box model. SB carried out the scientific analysis of the aircraft data and drafted the manuscript with contributions from all co-authors.

**Acknowledgments, Samples, and Data**

This work was supported by National Science Foundation (NSF 9188-1524) and the National Institute of Standards and Technology (NIST). The authors are grateful for the flight crew and the many scientists who helped to collect ARIAs and

A$^2$BC observations. The ARIAs flight data are available at: https://www-air.larc.nasa.gov/cgi-bin/ArcView/korusaq?OTHER=1#top. Background global map is from Esri, available at: https://www.arcgis.com/home/item.html?id=21b4ba14d9e5472d97afcbb819f7368e#overview. Additionally, we thank Glenn Wolfe of NASA for his support of the F0AM box modelling effort and Gabriele Pfister and Frank Flocke at NCAR for providing helpful discussion during manuscript preparation. The scientific results and conclusions, as well as any views or

opinions expressed herein, are those of the authors and do not necessarily reflect the views of NIST, NSF, or NOAA.

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

**Table 1. Y-12 research aircraft instrumentation during ARIAs.**

| Variable | Method |
|---|---|
| Aircraft Position | Global Positioning System (GPS) |
| Meteorology (Temperature, Relative humidity, Pressure, 2-D Wind) | Cloud water inertial probe (CWIP) |
| Greenhouse Gases ($CO_2$/$CH_4$/CO/$H_2O$) | Cavity Ring Down Spectroscopy Picarro Model G24201-m |
| Ozone ($O_3$) | UV-absorption, TECO 49C |
| Sulfur dioxide ($SO_2$) | Pulsed fluorescence, TECO 43C |
| Nitrogen dioxide ($NO_2$) | Cavity enhanced absorption spectroscopy, Los Gatos RMT-200 CRDS |
| NO/$NO_y$ | Chemiluminescence, modified TECO 42C with an external Molybdenum converter at 375°C |
| Aerosol Scattering, $b_{scat}$ (450, 500, 700, nm) | Nephelometer, TSI Model 3563 |
| Aerosol Absorption, $b_{abs}$ (565 nm) | Particle Soot Absorption Photometer (PSAP) |
| Black Carbon (370, 470, 520, 590, 660, 880, 950 nm) | Aethalometer, Magee Model AE31 |
| Black Carbon | Single-Particle Soot Photometer (SP2) |
| VOCs | Grab Canisters, GC-MSD/FID |

**Table 2. Aircraft monitoring results (all altitudes mostly in the PBL, 1-second data) in comparison with other airborne studies in the region. All units ppbv.**

| | This Study, ARIAs | | KORUS-AQ[*] | Yellow Sea, coastal and offshore[a] | | YRD[b] | Northeast China[c] | Northern China[d] | Bohai Sea[e] | Japan Sea[f] |
|---|---|---|---|---|---|---|---|---|---|---|
| | May-June 2016 | | May 24-29 | April 2011 | | October 2007 | April 5, 2007 | April 2006 | March 2002 | April 1996 |
| ppbv | Ave (STD) | Min/Max | Ave. (STD) | Ave | Min/Max | Min/Max | Ahead of cold front | Ave | Min/Max | Min/Max |
| $NO_x$ | 5.1 (7.9) | 0/53.2 | 1.3 (4.9) | 2.45 | 0.49/9.58 | 3/40 | - | 5.01 | -/18 | -/- |
| CO | 290.7 (309.6) | 80.5/6054 | 258.2 (144.5) | 980 | 630/1950 | 3000/7000 | ~300 | - | -/- | -/- |
| $O_3$ | 85.0 (15.6) | 45.0/145.6 | 89.8 (17.5) | 76.3 | 43.0/126.5 | 20/60 | ~70 | 43.8 | 35/65 | 70/90 |

[*] Statistics calculated for 1-second data during three flights at all altitudes during the "extreme pollution period" (Choi et al., 2019) where the KORUS-AQ DC-8 flew over the Yellow Sea to measure outflow from China.
[a] Yang et al., (2016).
[b] Geng et al., (2009).
[c] Dickerson et al., (2007)
[d] Wang et al., (2008).
[e] Hatakeyama et al., (2005).
[f] Inomata et al., (2006).

**Table 3.** Comparison of $CO/CO_2$ ratios during ARIAs to other ground-based and aloft measurements in China and developed regions of the world.

| Study | Location | Year | $CO/CO_2$ (%) |
|---|---|---|---|
| This Study* | North China Plain | May-June 2016 | 3.1 |
| Wang et al., 2010 | Miyuan, rural Beijing | Winter 2004 | 5.8 |
| | | Winter 2008 | 3.8 |
| Huang et al., 2015 | Nanjing, China | 2011 | 3.4-4.2 |
| Silva et al., 2013 | Space-based Megacities | June 2009-May 2010 | Beijing/Tianjin: 4.3 London: 0.6 Mumbai: 1.4 New York: 1.3 |
| Han et al., 2009 | Beijing, China | 2005-2006 | Fall: 3.0 Winter: 4.4 |
| Tang et al., 2018* | West Sea | May-June 2016 | 2.8 |
| | Seoul | | 0.9 |

| | Xia et al., 2020 | Jingdezhen station, central China, airflow from N China | December 2017 | 2.6 |
| | | Jingdezhen station, airflow from SW China | 18-21 January 2017 | 1.4 |
| | Ren et al., 2018 | Baltimore/Washington, D.C. | Winter 2016 | 0.53 |

*=Aircraft studies

**Table 4. Comparison of the top 10 most abundant species measured in this study with other ground observations in China (Units:**

 **ppbv).**

| | This Study | | | 43 Cities[a] | QZ[b] | GZ[d] | FS[e] | LZ[f] | BJ[b] | N. |
|---|---|---|---|---|---|---|---|---|---|---|
| | May-June 2016 | | | January-February 2001 | July 2014 | June 2011-May 2012 | December 2008 | June 2013-August 2013 | May 2014 | Ju Au 20 |
| | Ave | % | Range | Range | Ave | Ave | Ave | Ave | Ave | Ra |
| Ethane | 2.65 | 17.0 | 1.80-4.15 | 3.7-17.0 | 3.53 | 3.66 | 16.91 | - | 4.37 | 0.( |
| Propane | 1.39 | 8.9 | 0.98-1.89 | 1.5-20.8 | 1.31 | 4.34 | 16.26 | 3.40 | 2.44 | 0.: |
| Ethylene | 0.88 | 5.7 | 0.18-3.54 | 2.1-34.8 | 1.92 | 2.99 | 28.46 | - | 2.33 | 0.: |
| Acetylene | 0.80 | 5.1 | 0.23-1.93 | 2.9-58.3 | 1.94 | - | 32.82 | - | 2.17 | 0.( |
| Toluene | 0.76 | 4.9 | 0.03-4.40 | 0.4-11.2 | 0.48 | 4.59 | 18.87 | 1.01 | 1.33 | 0.( |
| i-Pentane | 0.67 | 4.3 | 0.03-5.44 | 0.3-18.8 | 0.60 | 1.72 | 1.84 | 2.43 | 0.99 | 0.( |
| i-Butane | 0.62 | 4.0 | 0.06-3.96 | 0.4-4.6 | - | 2.67 | 4.66 | 2.43 | 1.03 | 0.( |
| Benzene | 0.51 | 3.3 | 0.06-2.18 | 0.7-10.4 | 0.81 | 0.62 | 6.00 | 1.94 | 0.82 | 0.( |
| 2,2,4-Trimethylpentane | 0.43 | 2.8 | 0.01-5.42 | - | - | 0.22 | - | 0.10 | - | 0.( |
| 2-Methylheptane | 0.40 | 2.6 | 0.01-5.52 | - | - | 0.08 | 0 | 1.49 | - | 0.( |

[a]43 Cities, China (Barletta et al., 2005).
[b]QZ, Quzhou, BJ, Beijing (Li et al., 2015).
[c]NJ, Nanjing, Yangtze River Delta (An et al., 2017).
 [d]GZ, Guangzhou, Pearl River Delta (Zou et al., 2015).
[e]FS, Foshan, Pearl River Delta, haze days (Guo et al., 2011).
[f]LZ, Lanzhou (Jia et al., 2016).

**Table 5. Top 10 VOC species (mean and percentage breakdown) which contribute to ozone formation based on OH reactivity and Ozone formation potential during ARIAs.**

| OH Reactivity | | | Ozone Formation Potential | | |
|---|---|---|---|---|---|
| Species | Mean ($s^{-1}$) | % | Species | Mean (ppbv $O_3$) | % |
| Ethylene | 0.18 | 15.7 | Toluene | 5.81 | 19.6 |
| Propylene | 0.11 | 9.6 | Ethylene | 4.65 | 15.7 |
| Toluene | 0.10 | 8.9 | m/p-Xylene | 1.87 | 6.3 |
| 2-Methylheptane | 0.09 | 7.9 | Propylene | 1.72 | 5.8 |
| Isoprene | 0.08 | 7.2 | i-Pentane | 1.47 | 5.0 |
| i-Pentane | 0.06 | 5.1 | 2,2,4-Trimethylpentane | 1.30 | 4.4 |
| m/p-Xylene | 0.05 | 4.1 | 2-Methylheptane | 1.02 | 3.4 |
| 2,3,4-Trimethylpentane | 0.03 | 3.0 | i-Butane | 0.93 | 3.1 |
| 2,2,4-Trimethylpentane | 0.03 | 2.8 | o-Xylene | 0.72 | 2.4 |
| Propane | 0.03 | 2.7 | 1,2,4-Trimethylbenzene | 0.67 | 2.3 |

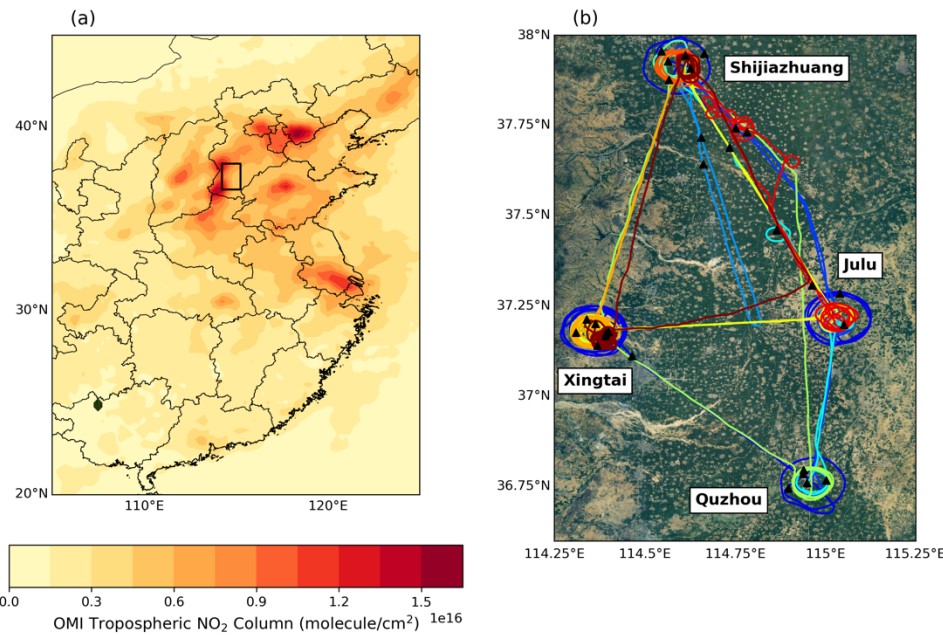

**Figure 1. (a) May and June 2016 OMI tropospheric column NO₂ from NASA Goddard Earth Sciences Data and Information Services Centre. The North China Plain is clearly seen in the centre with high column NO₂ concentrations; the black rectangle indicates ARIAs campaign domain and corresponds to the region shown in panel b. (b) Map of 11 ARIAs flight tracks (colored by flight number) and location of VOC samples (black triangles). The background map is provided by Esri, Copyright: ©2009.**

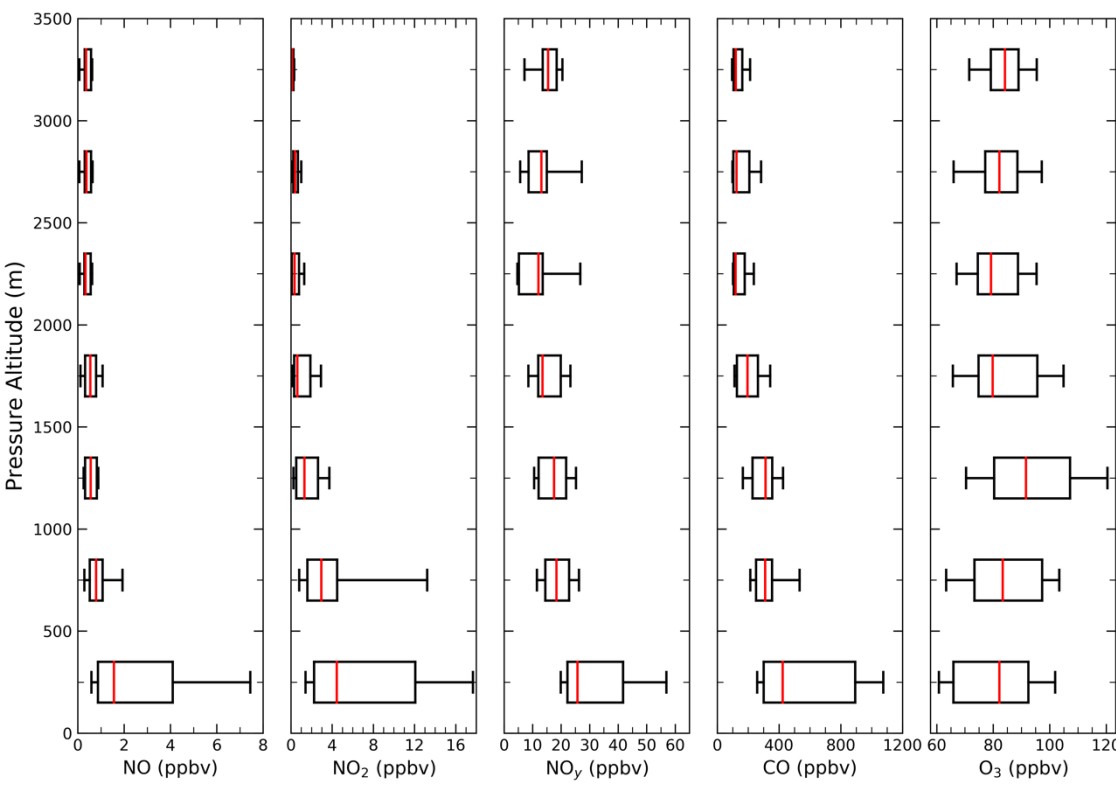

**Figure 2. Box and whisker plots of 1-second profiles of NO, NO₂, NOᵧ, CO, and O₃ for data collected in 500 m bins. The whiskers show the 10th and 90th percentiles, the box denotes the 25th and 75th percentiles, and the central red line indicates the median value within each bin. Average PBL height for all ARIAs flight is ~1500 m. The total number of observations at altitudes above 2500 m of NO and NOᵧ is small (~2,200 or about ~30 minutes of measurements) since the NO/NOᵧ instrument cannot measure both species simultaneously.**

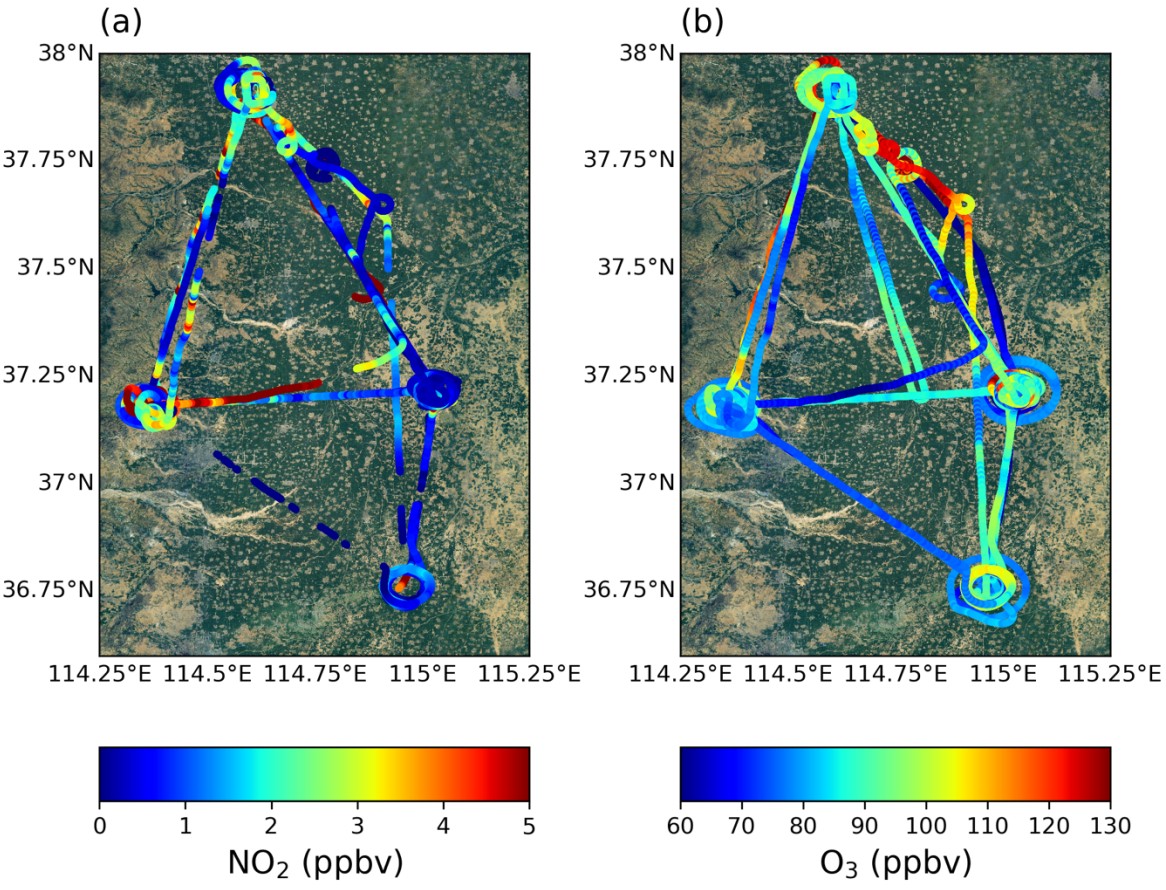

**Figure 3.** Maps of the ARIAs flight track colored by the 1-second measured mixing ratio (ppbv) of $NO_2$ (a) and $O_3$ (b). The background map is provided by Esri, Copyright: ©2009.

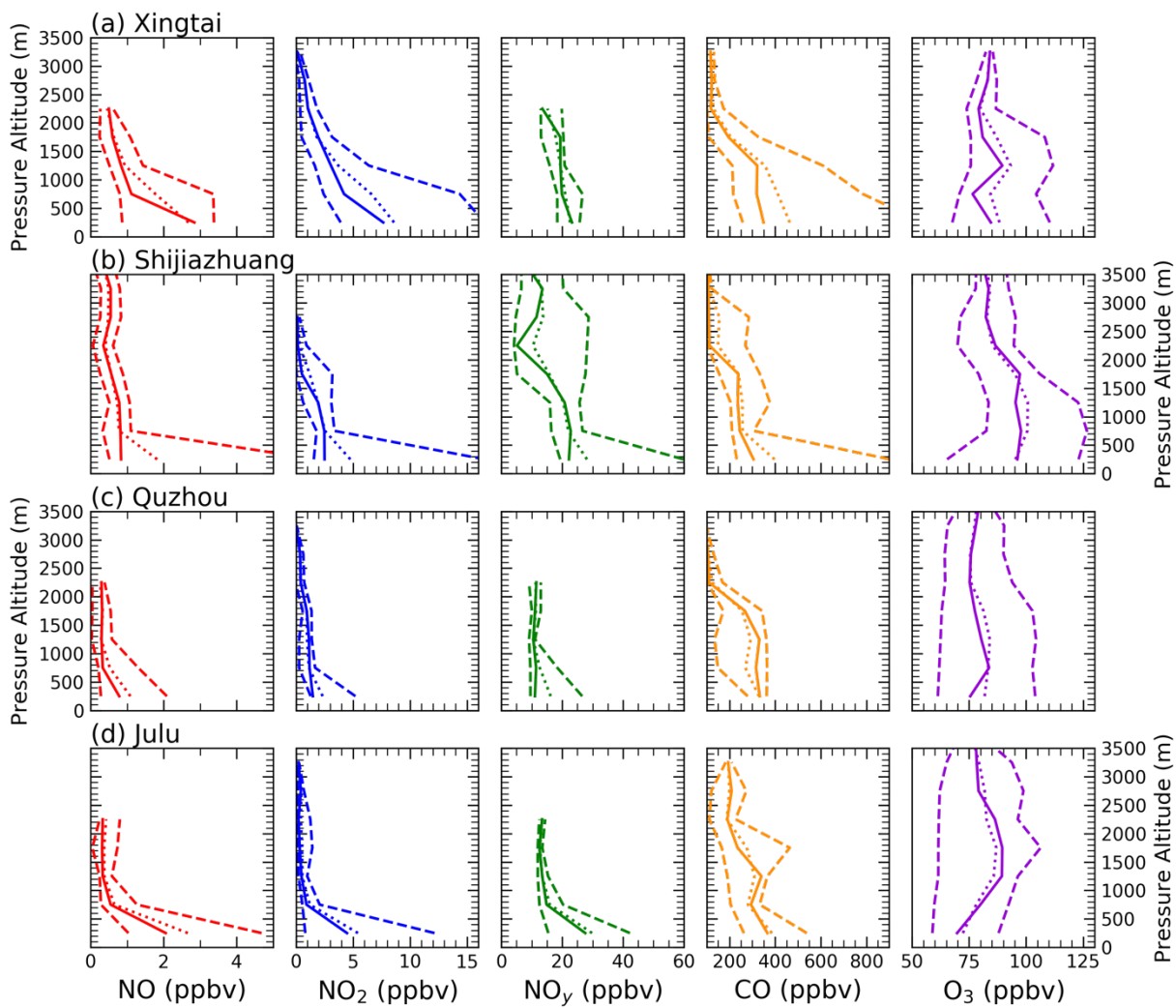

**Figure 4. Vertical profiles of 1-second NO (red), NO₂ (blue), NOᵧ (green), CO (orange), and O₃ (purple) in 500 m bins over the 4 spiral locations: Xingtai (a), Shijiazhuang (b), Quzhou (c), and Julu (d). The dashed lines indicate the 10th and 90th percentiles, the solid line is the median and the dotted line is the mean. We remove observations of NO/NOy above 2500 m over three spiral locations due to limited measurements.**

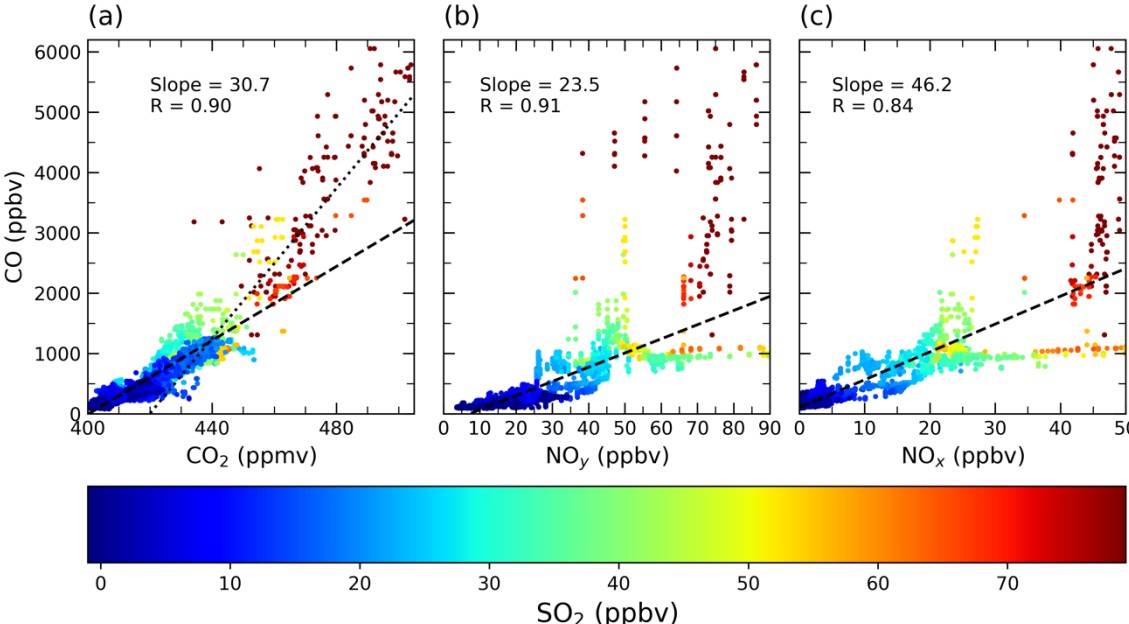

**Figure 5.** Scatter plots of 1-second (a) CO and $CO_2$, (b) CO and $NO_y$, and (c) CO and $NO_x$ colored by the $SO_2$ mixing ratio for all ARIAs flights. The dashed line shows the linear regression for each plot. The dotted line in panel a indicates the higher ratio commonly associated with biomass and biofuel burning.

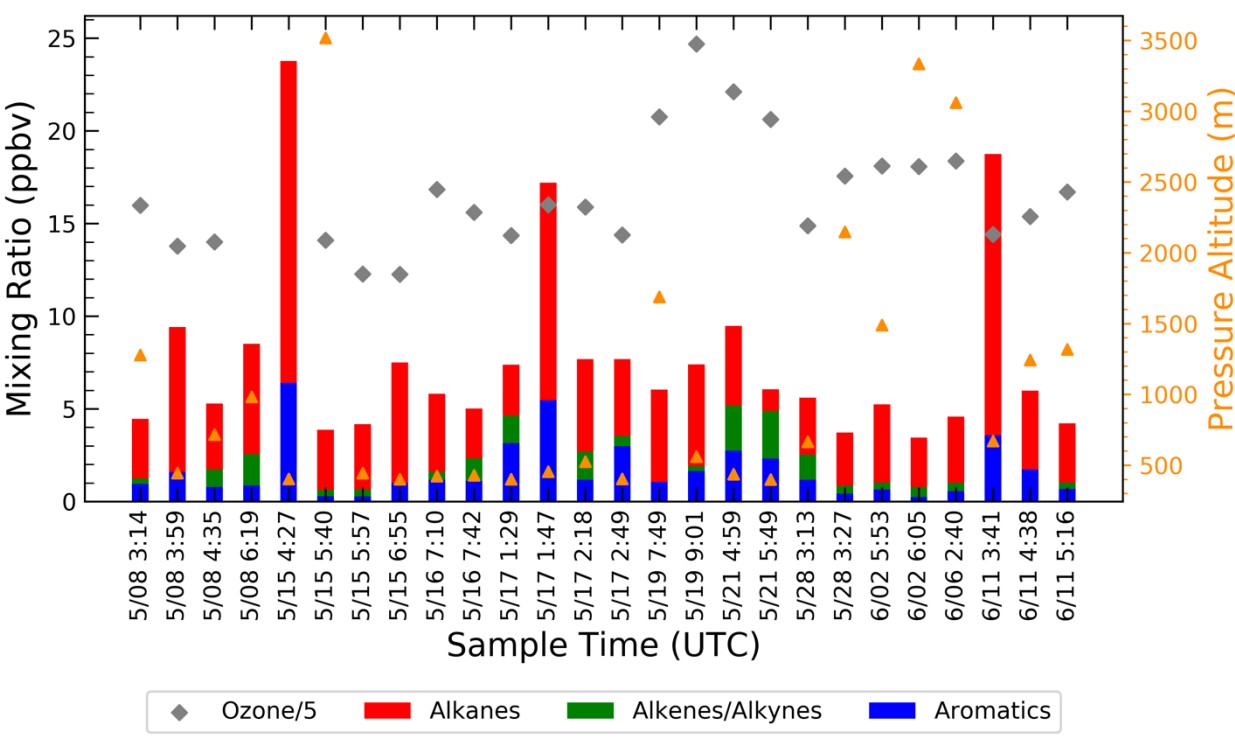

**Figure 6. Total VOC mixing ratio for each WAS canister during ARIAs colored by alkanes (red), alkenes/alkynes (green), and aromatics (blue). The concurrent O₃ mixing ratio, divided by 5, is shown in grey diamonds (using left y-axis) and the pressure altitude of the sample is denoted by orange triangles (using right y-axis).**

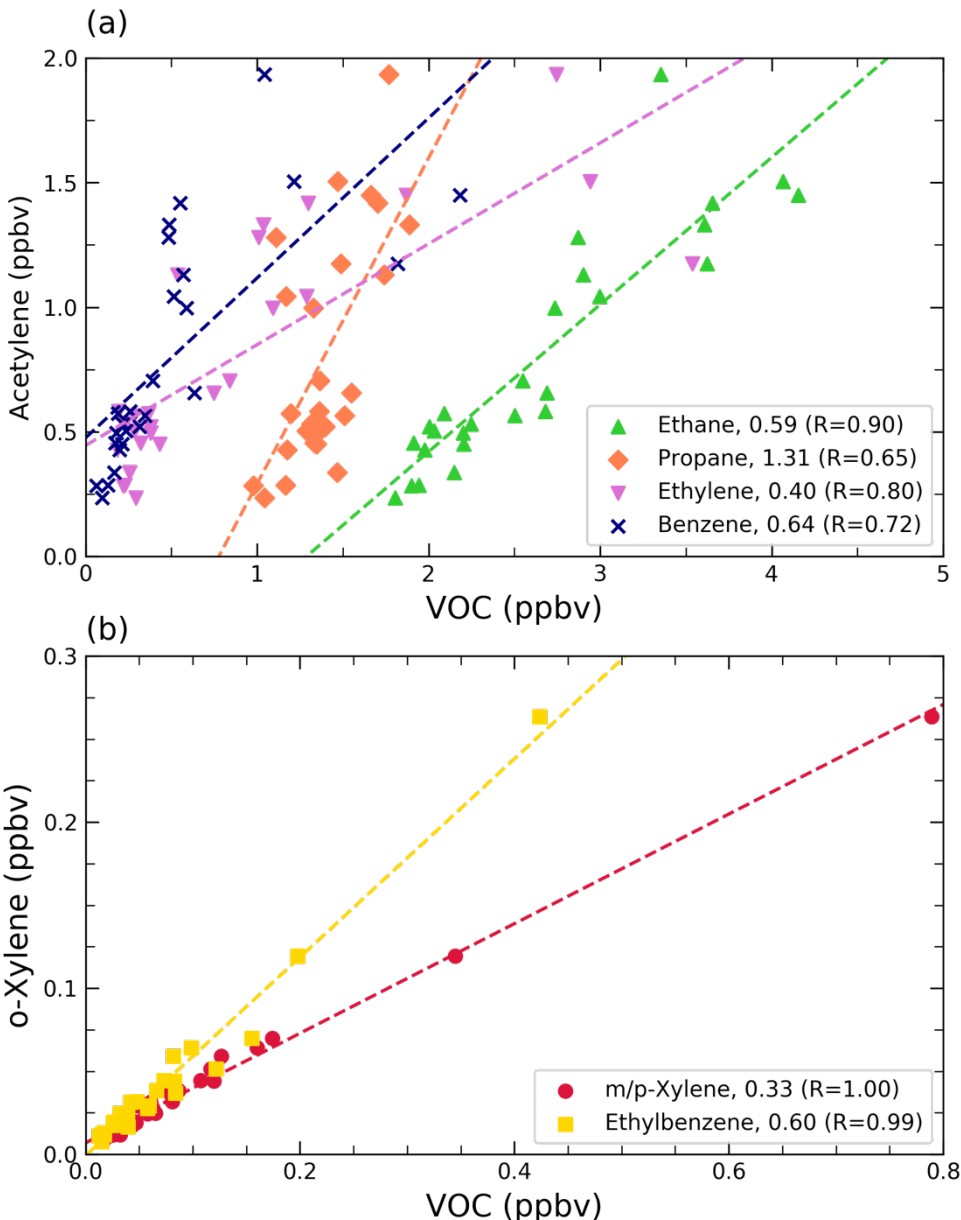

**Figure 7. (a) Scatterplot of acetylene with ethane (green upward triangles), propane (orange diamonds), ethylene (purple downward triangles), and benzene (navy ×'s) for all WAS canisters during ARIAs. (b) Regression plots of o-xylene with m/p-xylene (red circles) and ethylbenzene (yellow squares). The dashed lines show the results of a linear least squares regression line for all data points.**

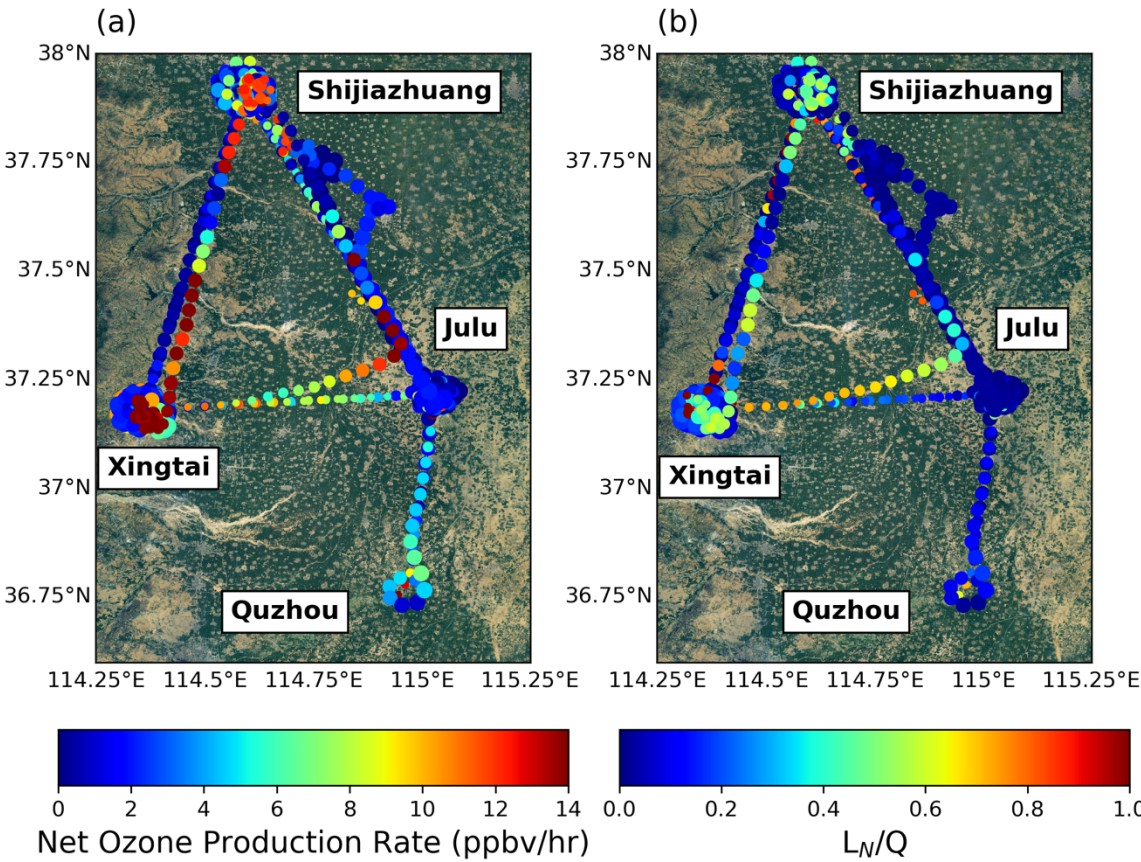

**Figure 8. (a) Map of the net production rates of O₃ calculated using F0AM box model results along the Y-12 flight track during**
**ARIAs. (b) Map of L_N/Q, an O₃ sensitivity indicator, along the Y-12 flight path. Ozone production is VOC-sensitive when L_N/Q >0.5**
**and NO_x-sensitive when L_N/Q <0.5 (Kleinman, 2005a). The size of the dots in both plots is proportional to the production rate of O₃.**
**The background map is provided by Esri, Copyright: ©2009.**

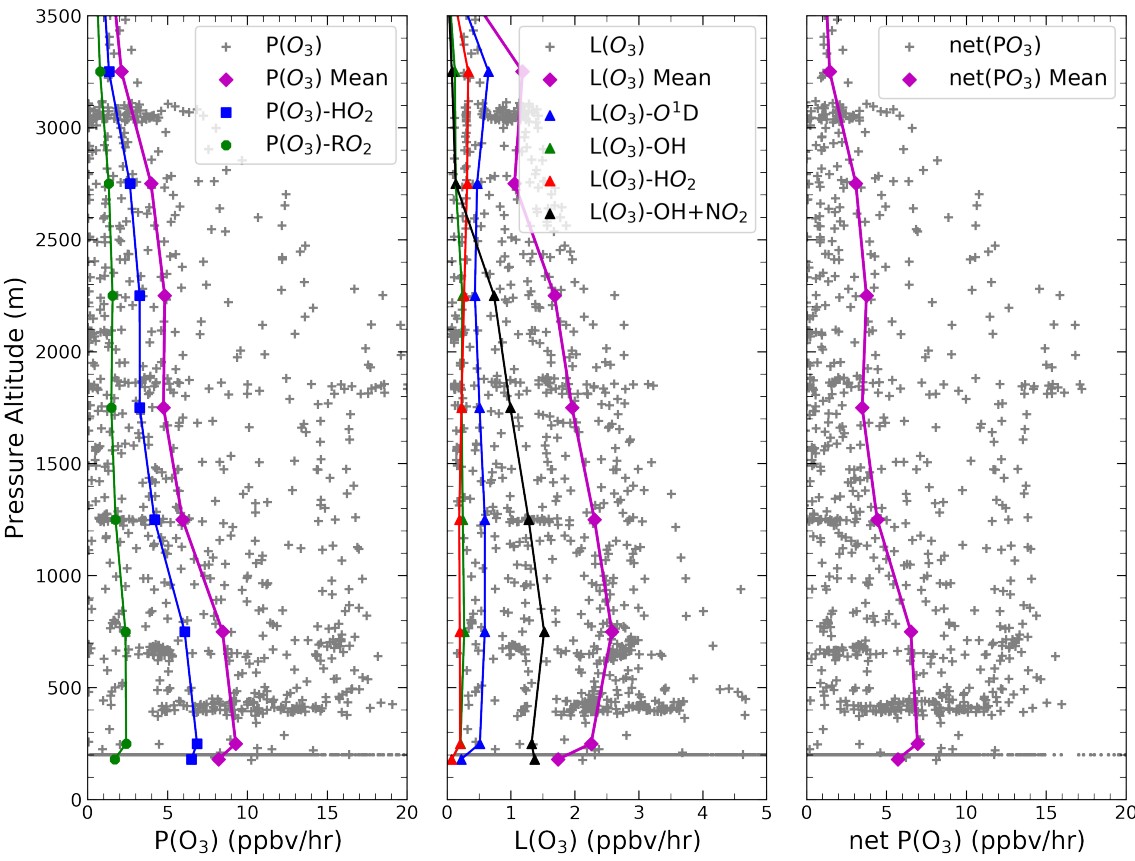

**Figure 9. Vertical profiles of the rate of production of O₃ (left), O₃ loss rate (middle), and net O₃ production rate (right) during ARIAs.**