# Peer review of "Measurement Report: Aircraft Observations of Ozone, Nitrogen Oxides, and Volatile Organic Compounds over Hebei Province, China"

_Atmospheric Chemistry and Physics, 2020_

## Referee Comment (RC1) · Anonymous Referee #1 · 15 Jul 2020

Review of "Measurement Report: Aircraft Observations of Ozone, Nitrogen Oxides, and Volatile Organic Compounds over Hebei Province, China"

A critical evaluation and assessment of what worked and what did not during the ARIAs campaign is missing in the present manuscript.

CO /CO2 ratio: How do the measurements of CO / CO2 compare to ground based measurements in urban centers of China? How do the CO / CO2 of the ARIAs study compare to measurements on other continents where pollution control measures have led to decreasing CO / CO2 ratios over time? How do the CO / CO2 ratios in plumes that are associated with biomass burning (Fig. 5a) compare to studies of biomass burn-
ing emission ratios? How many flights showed evidence of biomass burning emissions such as from past harvest residue burns?

Hydrocarbon profiles: How do the hydrocarbon values and their enhancement ratios to CO measured during ARIAs compare to ground based measurements in metropolitan areas of China, Europe or the US? How do they compare to biomass burning profiles?

Fig. 4: How many vertical profiles were flown over each of the four cities? The uniformly high NOy values from 0 to 3 km altitude over 3 of the cities are puzzling. In particular the uniformly high NOy values above 2300 m are in contrast to cleaner conditions at these altitudes as indicated by the CO mixing ratios. In contrast, over the home airport near Shijiazhuang the NOy measurements show a much wider range of mixing ratios throughout the altitude range of the flights. How consistent were the NOy measurements throughout the deployment?

The instantaneous O3 production rate and the VOC or NOx limitation: As intense city or power plant plumes age and mixing with the surrounding air during transport, the photochemical ozone production tends to transition from being more VOC to more NOx limited. To capture these transitions and adjustments with a photochemical box model is challenging and is not captured by running the box model simply for 3 days as done in the present paper (line 143). How was the photochemical box model run for the present study? Were the j-values held constant or was their diurnal cycle taken into account?

---

## Referee Comment (RC2) · Anonymous Referee #2 · 24 Jul 2020

This work presents aircraft measurements of O3 and its precursors in Hebei Province, China, aiming at understanding the production of ozone within the planetary boundary layer (PBL). They presented vertical profiles of trace gas species, including O3, NOx, CO, and VOCs. A box model was used to relate those concentrations to the O3 production rate and to assess the O3 production and OH reactivity relevant to the VOC/NOx ratio. Their analysis showed that measured O3 levels ranged from 52 to 142 ppbv, with the peak median concentration ($\sim$94 ppbv) between 1000 and 1500 m. The NOx concentrations exhibited strong spatial and altitudinal variations, ranging from 0.15 to 49 ppbv. They presented the ratios of CO/NOy and CO/CO2 to indicate the prevalence of low efficiency combustion from biomass burning and residential coal burning. Their

measurements of concentrations of total measured VOCs showed that alkanes and alkenes/alkynes were responsible for 74% of the total VOC reactivity, while aromatics contributed the most to the total Ozone Formation Potential (43%) with toluene, m/p-xylene, ethylene, propylene, and i-pentane playing significant roles in the production of O3 in this region. Their box model calculations constrained by measured precursors indicated the peak rate of mean O3 production was ~7 ppbv/hour below 500 m. They also showed that pollution frequently extended above the PBL into the lower free troposphere, where NO2 mixing ratios (~400 pptv) led to net O3 production rates up to ~3 ppbv/hour and this pollution traveled extended distances downwind. They concluded the O3 sensitivity regime as NOx-limited throughout the PBL, while VOC-limited at low altitudes near urban areas. Overall, there are very limited measurements on the vertical profiles of ozone and its precursors as well as an assessment of vertical ozone production and OH reactivity in this region. As such, this work is publishable in ACP, after the following issues have been adequately addressed. Main points (1) I am surprised that they did not found much contribution from biogenic VOCs between May and June 2016 in this region. Ground-based measurements in NCP have clearly showed a role of BVOCs in ozone and PM production (Wang et al., Use of a mobile laboratory to evaluate changes in on-road air pollutants during the Beijing 2008 summer Olympics, Atmos. Chem. Phys. 9, 8247, 2009; Guo et al., Elucidating severe urban haze formation in China, Proc. Natl. Acad. Sci. USA 111, 17373, 2014). Some comparison with ground-based measurements and discussions of the contribution of BVOCs to ozone production would be essential. (2) I would also think that the vertical profile in ozone production within the PBL also reflects photochemistry, which is closely related to PBL height and PM levels. Specifically, it has been known that the PBL height strongly regulates the photolysis rate (O1D) and there exists a strong feedback between PBL and PM (An et al., Severe haze in Northern China: A synergy of anthropogenic emissions and atmospheric processes, Proc. Natl. Acad. Sci. USA 116, 8657, 2019; Wu et al., Aerosol–photolysis interaction reduces particulate matter during wintertime haze events, Proc. Natl. Acad. Sci. USA 117, 9755, 2020). Typically, the trends in surface O3 and PM are believed to be anti-correlated. To what extend the PBL-photolysis interaction would impact their assessments of the vertical ozone production and OH reactivity in the present work? (3) Their measurements were made between May and June 2016. Recent studies have shown significantly different trends in O3 and PM precursors (particularly in NOx) in this region (Zhang et al., An unexpected catalyst dominates formation and radiative forcing of regional haze, Proc. Natl. Acad. Sci. USA 117, 3960, 2020). How did their measurements fit into those of trends for O3 and PM precursors? Minor points In general, the paper was reasonably-well written, but could be further improved to increase its readability. Below are a few examples. (1) The usage between past and presented tenses was interchangeable, but should be made consistent throughout the manuscript. "This analysis shows measured O3 levels ranged from ... The NOx concentrations exhibited ... Ratios of CO/NOy and CO/CO2 indicate . . ." (2) The phrase "26 whole air canisters" in the abstract was confusing. (3) The phrase "we see evidence of" in the abstract was rather causal. (4) The sentence "demonstrating both VOCs and NOx need further control to reduce aloft O3" in the abstract needs to be re-written.

---

## Author Comment (AC3) · 26 Aug 2020

Please see the attached pdf files for the revised manuscript and supplemental information.

Please also note the supplement to this comment:
https://acp.copernicus.org/preprints/acp-2020-194/acp-2020-194-AC3-supplement.zip

---

## Author Comment (AC4) · 26 Aug 2020

Please see the attached pdf files for the revised manuscript and supplemental information.

Please also note the supplement to this comment:
https://acp.copernicus.org/preprints/acp-2020-194/acp-2020-194-AC4-supplement.zip
* * *

---

## Author Response (AR1)

**Response to RC1**

We thank the anonymous referee for the thoughtful comments that has resulted in changes that improved the quality of our manuscript. We provide responses to the referee comments (in **bold**) below and provide the additional citations at the end of this response:

**A critical evaluation and assessment of what worked and what did not during the ARIAs campaign is missing in the present manuscript.**

Upon revision we will add details about the goals of the ARIAs campaign and what was actually accomplished. In particular, the following text will be added towards the end of the Introduction of the revised manuscript:

> The ARIAs campaign was designed to characterize and quantify the composition of trace gases and aerosol optical properties over Hebei to improve tools used to evaluate the effectiveness of air pollution reduction policies. Since air pollution transport from Asia typically peaks in early to mid-spring (Liu et al., 2003), we hoped to provide detailed altitude profiles over the Asian source region to enable Lagrangian experiments with KORUS-AQ, but only two sustained transport events occurred (Peterson et al., 2019). Despite the infrequent transboundary pollution events, ARIAs observations generated valuable characteristic pollution signatures that helped describe combustion efficiency and its impact downwind (Halliday et al., 2019), to correct model biases of CO in global chemistry-climate models (Gaubert et al., 2020) and to show that MOPITT bias increases at high CO concentrations (Tang et al., 2020). Furthermore, ARIAs measurements characterized aerosol optical properties in the planetary boundary layer and free troposphere during clean and polluted conditions (Wang et al., 2018), as well as used in the validation of MAX-DOAS profiles of $NO_2$, $SO_2$, HONO, HCHO, CHOCHO, and aerosols (Wang et al., 2019b).

**CO/CO2 ratio: How do the measurements of CO/CO2 compare to ground based measurements in urban centers of China? How do the CO/CO2 of the ARIAs study compare to measurements on other continents where pollution control measures have led to decreasing CO/CO2 ratios over time? How do the CO/CO2 ratios in plumes that are associated with biomass burning (Fig. 5a) compare to studies of biomass burning emission ratios? How many flights showed evidence of biomass burning emissions such as from past harvest residue burns?**

We agree that it is important to investigate the $CO/CO_2$ ratios from ground-based observations. However, the ARIAs campaign did not include GHGs measurements at ground stations. We have to rely on the ratios from literature and the KORUS-AQ campaign to answer the referee's concern. If the referee is aware of available $CO/CO_2$ ratios from Hebei, a reference would be appreciated. We will add the table below to the main text comparing our ARIAs $CO/CO_2$ ratio to the literature and discuss these studies with the following new text to be added to Section 3.1:

These measurements are illustrative of low-efficiency fossil fuel combustion, likely from residential coal burning as these observations were all collected at ~500 m, and are compared to other studies in Table 3. Our results indicating the prevalence of low-efficiency combustion agree with KORUS-AQ airborne data over the West Sea with 2.8% $CO/CO_2$ (Tang et al., 2018), as well as with December 2017 surface measurements at Jingdezhen station in central China of 2.6% when air mass transport was from northern China (Xia et al., 2020). Compared to earlier studies in rural and urban areas of Beijing in the mid-2000s (Han et al., 2009; Wang et al., 2010b) and to 2011 measurements in Nanjing (Huang et al., 2015), the ARIAs $CO/CO_2$ ratio is 0.1-2.7% lower, evident of some success of regional pollution control strategies. By contrast, our $CO/CO_2$ ratio is higher than satellite-derived ratios over megacities that have implemented extensive pollution control measures (Silva et al., 2013). Similarly, compared to airborne measurements from the 2015 Wintertime INvestigation of Transport, Emissions, and Reactivity (WINTER) campaign in the Baltimore/Washington, D.C. region (Ren et al., 2018), our $CO/CO_2$ ratio is about a factor of 6 larger.

When we went back to the data to identify the number of flights which sampled biomass burning plumes, we realized the 1-minute average data was missing some of these short events. We will instead use the 1-second data in Figure 5 in the revised manuscript to better show these plumes. The ARIAs $CO/CO_2$ ratios measured in plumes associated with biomass burning is ~6%. This ratio is comparable to past studies evaluating emission ratios from wheat straw burning in Hebei (Cao et al., 2008). We identified three ARIAs flights which briefly sampled biomass burning plumes. We will add these additional details with the following revised text to Section 3.1:

Higher $CO/CO_2$ ratios (~6%) with less than 0.1 ppm $SO_2$, as seen briefly during three ARIAs flights, are more in line with emissions from burning of wheat straw in Hebei of ~6% (Cao et al., 2008), and other inefficient, biofuel combustion.

We will revise a sentence in the Conclusions to now read:

Ratios of $CO/CO_2$ indicate inefficient combustion from residential coal and biomass burning throughout the region, but have decreased in China since the early 2000s suggesting the implementation of successful pollution control strategies.

We will add the following new table, which will be *Table 3* of the revised manuscript, in reply to this comment:

| Study | Location | Year | CO/CO$_2$ (%) |
|---|---|---|---|
| This Study* | North China Plain | May-June 2016 | 3.1 |
| Wang et al., 2010 | Miyuan, rural Beijing | Winter 2004 | 5.8 |
|  |  | Winter 2008 | 3.8 |
| Huang et al., 2015 | Nanjing, China | 2011 | 3.4-4.2 |
| Silva et al., 2013 | Space-based Megacities | June 2009-May 2010 | Beijing/Tianjin: 4.3 Mumbai: 1.4 New York: 1.3 London: 0.6 |

| Han et al., 2009 | Beijing, China | 2005-2006 | Fall: 3.0
 Winter: 4.4 |
|---|---|---|---|
| Tang et al., 2018* | West Sea | May-June 2016 | 2.8 |
| | Seoul | | 0.9 |
| Xia et al., 2020 | Jingdezhen station, central China, airflow from N China | December 2017 | 2.6 |
| | Jingdezhen station, airflow from SW China | 18-21 January 2017 | 1.4 |
| Ren et al., 2018 | Baltimore/Washington, D.C. | Winter 2016 | 0.53 |

*=Airborne studies

**Hydrocarbon profiles: How do the hydrocarbon values and their enhancement ratios to CO measured during ARIAs compare to ground based measurements in metropolitan areas of China, Europe or the US? How do they compare to biomass burning profiles?**

High concentrations of anthropogenic VOCs measured during ARIAs suggest that our flights are close to local VOCs sources, however we find that very few VOCs species exhibit a strong correlation with CO. Since CO is a marker of combustion, the lack of correlation indicates the lack of common source signatures and/or some photochemical aging of the sampled airmasses. For these reasons, we plan to add another column to Table S2 reporting the VOC/CO ratio where R>0.50 for 13 VOCs. In general, hydrocarbon enhancement to CO during ARIAs are lower than other metropolitan ground-based studies.

The following new text will be added to Section 3.2:

> Since CO can be marker for anthropogenically emitted hydrocarbons, particularly combustion products, we first use the ratios of various VOCs to CO to reveal insight into changes in emissions in the region. Ratios of VOCs to CO can vary substantially among cities (Baker et al., 2008; Warneke et al., 2007), but in general can provide details about fuel types and combustion efficiency between metropolitan regions. Despite ARIAs measurements sampling in close proximity to local VOCs sources, most VOCs do not correlate strongly with CO, reflective of the lack of common source signatures and some photochemical aging of the sampled airmasses. We report slopes of VOCs/CO in Table S2 when R>0.50. Ethane has the strongest correlation with CO (R=0.72) and the slope (2.5 pptv/ppbv) agrees well with ratios from urban areas of the United States in 1999-2005 (2.4 pptv/ppbv) (Baker et al., 2008) as well as with charcoal burning emission ratios (Andreae and Merlet, 2001). The ARIAs emission ratio of benzene/CO (1.8 pptv/ppbv) is slightly higher than found in urban regions of the United States (0.7, Baker et al., 2008) and Mexico City (0.93-1.20, Apel et al., 2010), likely due to higher emissions by widespread combustion of coal and agricultural residues (Zhang et al., 2015). By contrast, the ARIAs emission ratios of ethylene and acetylene to CO (2.9. and 1.4 pptv/ppbv, respectively) are lower than observed in urban areas in the United States (4.1

and 3.4 pptv/ppbv, respectively) and Mexico City (7.90-8.40 and 8.20-9.60 pptv/ppbv, respectively), where the dominant source was reported to be transportation-related (Baker et al., 2008). The lower ratio of ethylene/CO is comparable to emission ratios reported from charcoal burning (2.3 pptv/ppbv) (Andreae and Merlet, 2001).

**Fig. 4: How many vertical profiles were flown over each of the four cities? The uniformly high NOy values from 0 to 3 km altitude over 3 of the cities are puzzling. In particular the uniformly high NOy values above 2300 m are in contrast to cleaner conditions at these altitudes as indicated by the CO mixing ratios. In contrast, over the home airport near Shijiazhuang the NOy measurements show a much wider range of mixing ratios throughout the altitude range of the flights. How consistent were the NOy measurements throughout the deployment?**

We thank the referee for bringing this question to our attention. There were 34 profiles over Shijiazhuang, 20 over Xingtai, 16 over Julu, and 7 over Quzhou. As stated in the caption of Figure 2, the total number of NO/NOy observations above 2500 m is small (~30 minutes total of measurements) since the instrument switched between NO and NOy at 10 s intervals and could not measure both species simultaneously. The NOy converter required lots of power, so we did not turn on the instrument on frequently.

The caption of Figure 4 denotes the data is 1-second observations, but the 1-minute version was used accidentally. For the 1-minute average data, Shijiazhuang has the most measurements above 2500 m (20-40 data points in each bin above this altitude) since we regularly conducted profiles at the beginning and end of each flight. The other spiral locations have less than 10 1-minute average data points (and usually less than 5 in two of the three locations). When the 1-second data is used, there is ~1500 data points in each bin over Shijiazhuang, while the other spiral locations generally have less than 400 data points. To avoid overinterpretation of the limited NO/NOy observations over the other three spiral locations, we will cut off the profiles at 2500 m for NO and NOy over Xingtai, Quzhou, and Julu.

We will add the following sentence to Section 2.1 of the revised manuscript:

> We remove observations of NO/NO$_y$ over three spiral locations due to limited measurements.

We have removed the following phrase after "Median profiles of NOy below 500m are highest over Julu (27.6 ppbv)" from Section 3.1 to avoid overinterpretation of the limited high altitude NO/NOy measurements:

> while aloft concentrations are similar between the spiral locations (~13 ppbv between 2500-3000 m).

Figure 4 will be updated as shown below, removing the high-altitude NOy measurements from the three spiral locations, with the updated figure caption reading:

Figure 4. Vertical profiles of 1-second NO (red), $NO_2$ (blue), $NO_y$ (green), CO (orange), and $O_3$ (purple) in 500 m bins over the 4 spiral locations: Xingtai (a), Shijiazhuang (b), Quzhou (c), and Julu (d). The dashed lines indicate the 10th and 90th percentiles, the solid line is the median and the dotted line is the mean. We remove observations of NO/NOy above 2500 m over three spiral locations due to limited measurements.

[Figure]

**The instantaneous O3 production rate and the VOC or NOx limitation: As intense city or power plant plumes age and mixing with the surrounding air during transport, the photochemical ozone production tends to transition from being more VOC to more NOx limited. To capture these transitions and adjustments with a photochemical box model is challenging and is not captured by running the box model simply for 3 days as done in the present paper (line 143). How was the photochemical box model run for the present study? Were the j-values held constant or was their diurnal cycle taken into account?**

We apologize for the confusion in the description of the box modeling simulations. The idea is that a spectrum of fresh and aged air parcels were observed and modeled. The box model was run for seven ARIAs flights. On the days that a flight occurred, a surface simulation was also run. The three days as referenced in line 143 of the original paper were intended to describe the number of days the model was run in solar cycle mode. In the solar cycle configuration, the model allows the solar zenith angle to evolve in "real time" over the course of a model step. Photolysis frequencies, not measured during ARIAs or at the $A^2BC$ supersite, evolve over the course of a model step and are calculated by combining cross sections and quantum yields with solar spectra derived from the NCAR Tropospheric Ultraviolet and Visible (TUV) version 5.2 radiation model. These agree within experimental error with direct measurements (Shetter et al., 2003). At the start of the model run, input solar zenith angle, altitude or elevation, $O_3$ column, and surface albedo are used for linear interpolation across TUV lookup tables (F0AM's "hybrid" method). We use SZA and altitude/elevation from ARIAs/$A^2BC$ measurements and constant values for ozone column (325 DU) and surface albedo (0.17), which we estimate based on concurrent data from the OMI level-3 OMDOAO3e data product. We have expanded the methods Section 2.2 to more clearly explain how the photochemical box model was run. The revised Section 2.2 text will read as follows:

[revised manuscript text omitted]

**Response to RC2**

We thank the anonymous referee for the thoughtful comments and remarks. We provide responses to the referee comments (in bold) below and include the additional references cited at the end of this response:

**This work presents aircraft measurements of O3 and its precursors in Hebei Province, China, aiming at understanding the production of ozone within the planetary boundary layer (PBL). They presented vertical profiles of trace gas species, including O3, NOx, CO, and VOCs. A box model was used to relate those concentrations to the O3 production rate and to assess the O3 production and OH reactivity relevant to the VOC/NOx ratio. Their analysis showed that measured O3 levels ranged from 52 to 142 ppbv, with the peak median concentration (∼94 ppbv) between 1000 and 1500 m. The NOx concentrations exhibited strong spatial and altitudinal variations, ranging from 0.15 to 49 ppbv. They presented the ratios of CO/NOy and CO/CO2 to indicate the prevalence of low efficiency combustion from biomass burning and residential coal burning. Their measurements of concentrations of total measured VOCs showed that alkanes and alkenes/alkynes were responsible for 74% of the total VOC reactivity, while aromatics contributed the most to the total Ozone Formation Potential (43%) with toluene, m/p- xylene, ethylene, propylene, and i-pentane playing significant roles in the production of O3 in this region. Their box model calculations constrained by measured precursors indicated the peak rate of mean O3 production was ∼7 ppbv/hour below 500 m. They also showed that pollution frequently extended above the PBL into the lower free tropo- sphere, where NO2 mixing ratios (∼400 pptv) led to net O3 production rates up to ∼3 ppbv/hour and this pollution traveled extended distances downwind. They concluded the O3 sensitivity regime as NOx-limited throughout the PBL, while VOC-limited at low altitudes near urban areas. Overall, there are very limited measurements on the vertical profiles of ozone and its precursors as well as an assessment of vertical ozone production and OH reactivity in this region. As such, this work is publishable in ACP, after the following issues have been adequately addressed.**

**Main points (1) I am surprised that they did not found much contribution from biogenic VOCs between May and June 2016 in this region. Ground-based measurements in NCP have clearly showed a role of BVOCs in ozone and PM production (Wang et al., Use of a mobile laboratory to evaluate changes in on-road air pollutants during the Beijing 2008 summer Olympics, Atmos. Chem. Phys. 9, 8247, 2009; Guo et al., Elucidating severe urban haze formation in China, Proc. Natl. Acad. Sci. USA 111, 17373, 2014). Some comparison with ground-based measurements and discussions of the contribution of BVOCs to ozone production would be essential.**

We thank the reviewer for providing the references for the role of BVOCs on ozone and PM production and agree a discussion of BVOCs should be included. We plan to add the following paragraph to the Introduction of the revised manuscript to address this comment, including citation of the studies suggested by the reviewer:

> Natural emissions are the largest source of VOCs globally and react more efficiently with OH than most anthropogenic compounds (Di Carlo et al., 2004), but exhibit a strong seasonal, diurnal, and spatial dependence (Li et al., 2013). Biogenic VOCs have been found to play a significant role in the formation of $O_3$ at the surface (Ma et al., 2019;

Zong et al., 2018) and throughout the boundary layer in the NCP (Wang et al., 2008), as well as influence production of $PM_{2.5}$ (Guo et al., 2014) and secondary organic aerosols (SOA) (Wu et al., 2020b). In particular, isoprene has been estimated to account for 27% of the total $O_3$ production in June 2010 in Beijing (Mo et al., 2018), suggesting the need to consider biogenic isoprene emissions in formulating $O_3$ control strategies. Quantifying the abundance of $NO_x$ and the suite of VOC chemicals throughout the lower troposphere is urgently needed to better understand the photochemistry of $O_3$ production in the NCP, which in turn will lead to the development of successful mitigation strategies.

In this new section noted above, we choose not to cite Wang et al. (2009) as the referee suggested since this paper seems to be on a different topic. Instead, we include a publication where Wang et al. (2009) was a coauthor (Mo et al. (2018)), as well as a paper by Q. Wang et al. (2008) in *Science of the Total Environment*, which modeled the impacts of biogenic emissions of VOCs and $NO_x$ on the formation of tropospheric ozone during summertime in eastern China.

In the present study, we were not able to quantify many prevalent biogenic VOCs, such as alpha and beta pinene and monoterpenes; however, we did quantify isoprene. We plan to add a comparison of ARIAs isoprene measurements to the literature in Section 3.2 to the revised manuscript. The new text will read as follows:

> Additionally, our observations have higher amounts of branched alkanes, such as 2,2,4-trimethylpentane and 2-methylheptane (both components of gasoline), but lower amounts of isoprene due to collection over mostly urban regions with lower ambient temperatures than the summer months. Since isoprene with a lifetime of hours (Seinfeld and Pandis, 2006) in the summer typically exhibits a strong vertical gradient in the PBL (Huang et al., 2017), we find the mean amount of isoprene measured during ARIAs is about 7 times lower than average May 2014 surface measurements in Beijing (Li et al., 2015), as well as ~200 pptv lower than June-July 2007 airborne measurements in the PBL in NE China (Xue et al., 2011).

Isoprene has been observed to be important near the surface but since ozone is made throughout the PBL, our observations expand the knowledge base for ozone formation. We add the following new text to Section 3.3.2 of the revised manuscript:

> At a surface site in Beijing (May 2014), Li et al. (2015) found m/p-xylene, ethylene, toluene, propylene, and o-xylene are most influential to OFP, while at a ground station in Tianjin (August 2018), Han et al. (2020) found that ethylene, isoprene, toluene, m/p-xylene, and propylene were important contributors to OFP. Our study supports a larger contribution of anthropogenic VOCs than biogenic VOCs in spring, although summer studies indicate a major role for isoprene to the formation of $O_3$ in the NCP (Han et al., 2020; Zong et al., 2018). Since isoprene is mostly emitted by biogenic sources during the warmer summer months with strong solar radiation and when soil moisture is sufficient for plant growth, we expect isoprene to have a larger impact on $O_3$ production in the summer than during spring, the time of our study.

Lastly, we add the following new sentence to the Conclusions:

In contrast to other surface studies in summer, we find a lower contribution of biogenic sources (e.g. isoprene) to the formation of $O_3$ in the PBL.

**(2) I would also think that the vertical profile in ozone production within the PBL also reflects photochemistry, which is closely related to PBL height and PM levels. Specifically, it has been known that the PBL height strongly regulates the photolysis rate (O1D) and there exists a strong feedback between PBL and PM (An et al., Severe haze in Northern China: A synergy of anthropogenic emissions and atmospheric processes, Proc. Natl. Acad. Sci. USA 116, 8657, 2019; Wu et al., Aerosol–photolysis interaction reduces particulate matter during wintertime haze events, Proc. Natl. Acad. Sci. USA 117, 9755, 2020). Typically, the trends in surface O3 and PM are believed to be anti-correlated. To what extend the PBL-photolysis interaction would impact their assessments of the vertical ozone production and OH reactivity in the present work?**

We agree that there are strong feedbacks between the PBL and PM and that the impact of aerosols on ozone production, even the sign of the effect, depends on their optical properties as well as vertical distribution (Dickerson et al., 1997; Kelley et al., 1995). The 0-D box model cannot simulate the depth of the PBL, but since our simulations were constrained by observations, automatically includes effects of dilution due to the height of the PBL. We will add the following revised text to the introduction as well as the recommended references to the manuscript:

> The role of VOCs on the formation of $O_3$ depends on the characteristics of the environment, including the main emission sources of primary pollutants and ambient temperature (Pusede et al., 2014), and the interaction of aerosols within the PBL to reduce photolysis (An et al., 2019). High aerosol concentrations have been shown to decrease photolysis and hinder summer surface $O_3$ formation by 25 ppbv on average in Xi'an, China (Feng et al., 2016), which pose a challenge for pollution control strategies.

The net impact of j(O3) and j(NO2) and thus the rate of ozone production was tested for in our model calculation, but is small. We also add new text to Section 2.2 describing the impact of aerosols on vertical ozone production:

> The impact of aerosols on $O_3$ production depends on the optical properties as well as the vertical distribution (Dickerson et al., 1997; Kelley et al., 1995). In the presence of scattering and absorbing aerosols, photolysis frequencies will be altered, thus changing the $O_3$ formation and atmospheric oxidizing capability (Wu et al., 2020a). Previous research over China has shown that as AOD increases, the extinction effect of aerosols on photolysis frequencies decreases due to a higher proportion of scattering aerosols under high AOD conditions (Wang et al., 2019a). Optical depth, single scattering albedo, and angstrom exponent during ARIAs (see Wang et al., 2018a) are used in the TUV online calculator (https://cprm.acom.ucar.edu/Models/TUV/Interactive_TUV/) to assess the impact of aerosols on photolysis frequencies. Most of the aerosol particles during ARIAs were concentrated in the lowest 2 km of the atmosphere with a single scattering albedo at 550 nm of 0.85 and an average AOD ~0.2. The impact of aerosol optical properties

measured during ARIAs on photolysis frequencies is small compared to the default setting, so no additional adjustments are made to the model values.

The OH reactivity calculation in this manuscript uses rate constants published by MCM and NIST, which represent optimal conditions in which there are no aerosols. We will add the following text to Section 3.3:

> In this section, we present results using the loss rate of each VOC species with OH and ozone formation potential (OFP) assuming no influence of aerosols. Since the aerosol effect on $O_3$ formation is dependent upon time of day (solar zenith angle), meteorology, levels of local and neighboring aerosols, and the VOC/NOx ratio, the calculations presented here are simplified compared to the more complicated chemical composition of the atmosphere, but are still useful to help inform control strategies.

We will add the following sentence to the conclusions to stress the importance of better understanding the aerosol impact of ozone production:

> The photochemistry of $O_3$ production is highly dependent upon the interaction of radiation and aerosols within the PBL and future work is needed to assess optical properties of aerosols at wavelengths relevant to photolysis of $O_3$ to $O(^1D)$ and thus OH.

**(3) Their measurements were made between May and June 2016. Recent studies have shown significantly different trends in O3 and PM precursors (particularly in NOx) in this region (Zhang et al., An unexpected catalyst dominates formation and radiative forcing of regional haze, Proc. Natl. Acad. Sci. USA 117, 3960, 2020). How did their measurements fit into those of trends for O3 and PM precursors?**

We agree with the referee that several recent studies have found different trends in O3 and PM precursors in this region and have added the following discussion at the end of the Introduction:

> The North China Plain is one of the most polluted regions in the world, but implementation of pollution reduction measures through the Five-Year Plans has allowed for decreasing trends of many pollutants. In particular, Zhang et al. (2020) found an increased number of days of clean/light haze and a decreased number of days with heavy haze, along with a significant decline of $SO_2$ concentrations. Similarly, using observations from MODIS and OMI, Si et al. (2019) found AOD and $SO_2$ to decrease from 2006 to 2015, while $NO_2$ rose by 4.79% in the NCP during this period. While surface $NO_2$ decreased 20% from May 2014 to December 2018 throughout China, there are still a large number of measurement stations with increasing trends of $NO_2$ due to changes in meteorological conditions and aerosol emissions (Fan et al., 2020), illustrating the need for more research characterizing air pollution in this region.

Since our measurements are limited to a small part of Hebei Province from 11 research flights in May and June 2016, we compare our aloft observations to surface ozone concentrations from the $A^2BC$ site located in Xingtai. We will add the following new text to Section 3.1 of the revised manuscript with the following figure, which will be *Figure S3,* included in the supplement:

The vertical profiles of $O_3$ compared to concurrent surface measurements in Xingtai indicates the $A^2BC$ site usually observed larger average concentrations than observed aloft, but this difference was highly dependent upon time of day (Fig. S3). The early afternoon profiles on May 8 showed average surface concentrations only slightly higher than the Y-12 measurements at ~400 m, while the mid-afternoon profiles on May 21 showed ~25 ppbv higher surface $O_3$ concentrations than Y-12 observations. At low altitudes (~700 m), the late morning flight (around 11:00 LST) on May 28 observed levels of $O_3$ ranging from 72-80 ppbv, comparable to average surface concentrations of 78 ppbv at the same time. By contrast, the afternoon flight (approximately 17:00 LST) at the same altitude later that day observed ~25 ppbv lower levels of $O_3$ compared to the surface (average=121 ppbv). All profiles on June 11 showed 10-30 ppbv lower average surface concentrations than measured during the Y-12 spirals.

Figure S3. Vertical profiles (N=20) of 1-second $O_3$ concentrations (ppbv) from the Y-12 (circles) compared to concurrent average concentrations measured at the $A^2BC$ site in Xingtai (diamonds). The average surface $O_3$ concentration was computed by averaging the 5-minute data interval starting 30 minutes before the spiral until 30 minutes after the spiral was completed.

[Figure]

**Minor points: In general, the paper was reasonably-well written, but could be further improved to increase its readability. Below are a few examples. (1) The usage between past and presented tenses was interchangeable, but should be made consistent throughout the**

**manuscript. "This analysis shows measured O3 levels ranged from ... The NOx concentrations exhibited ... Ratios of CO/NOy and CO/CO2 indicate . . ."**

We thank the reviewer for these readability suggestions. The present tense was used for the presentation of the analysis, while the past tense was used when discussing the airborne observations. The idea is to distinguish measurements (past tense) from general conclusions (present tense), i.e., concentrations were X implications are Y. We have decided to keep this style in the manuscript.

**(2) The phrase "26 whole air canisters" in the abstract was confusing.**

We will remove the phrase "26 whole air canisters" from the Abstract.

**(3) The phrase "we see evidence of" in the abstract was rather causal.**

We will revise this sentence to remove the casual phrase in the Abstract to now read:

> Ratios of $CO/CO_2$ indicate the prevalence of low efficiency combustion from biomass burning and residential coal burning, but indicate some success of regional pollution controls compared to earlier studies in China.

**(4) The sentence "demonstrating both VOCs and NOx need further control to reduce aloft O3" in the abstract needs to be re-written.**

We will change this phrase to read "demonstrating that control of both VOCs and NOx is needed to reduce aloft $O_3$ pollution over Hebei."

We also became aware of a recently published paper on air quality in China by Souri et al. (ACP, 2020). This paper focused on satellite observations and is an ideal complement to our airborne based analysis. Upon revision we propose to cite this paper in the following manner in Section 3.4:

[revised manuscript text omitted]

*Shetter, R. E., Junkermann, W., Swartz, W. H., Frost, G. J., Crawford, J. H., Lefer, B. L., Barrick, J. D., Hall, S. R., Hofzumahaus, A., Bais, A., Calvert, J. G., Cantrell, C. A., Madronich, S., Müller, M., Kraus, A., Monks, P. S., Edwards, G. D., McKenzie, R., Johnston, P., Schmitt, R., Griffioen, E., Krol, M., Kylling, A., Dickerson, R. R., Lloyd, S. A., Martin, T., Gardiner, B., Mayer, B., Pfister, G., Röth, E. P., Koepke, P., Ruggaber, A., Schwander, H. and van Weele, M.: Photolysis frequency of NO2: Measurement and modeling during the International Photolysis Frequency Measurement and Modeling Intercomparison (IPMMI), J. Geophys. Res. Atmos., 108(16), doi:10.1029/2002jd002932, 2003.

[revised manuscript text omitted]
. The first sample was collected on May 21 at 399 m pressure altitude. This sample was heavily polluted with i-butane (25.8 ppbv), i-pentane (57.7 ppbv), as well as longer chain alkanes like 2,3-dimethylbutane (4.2 ppbv), 2-methylpentane (5.9 ppbv), cyclopentane (2.7 ppbv), 2-methylheptane (16.1 ppbv), and 3-methylpentane (3.5 ppbv) in addition to aromatics like toluene (41.3 ppbv) and benzene (20.8 ppbv). Many of these compounds are typical of fuel evaporation or from petrochemical industries, indicating this canister may have directly sampled directly in the plume of one of these sources. Since this study is primarily focused with evaluating aloft VOCs away from their direct emission sources, the data from this canister were removed from this analysis.

The second contaminated sample was collected on May 28 at 3:36 UTC. This sample was filled to ambient pressure at 3000 m in relatively clean air, based on *in situ* observations at the time the canister was collected (CO=111 ppbv, $CH_4$=1890 ppbv, $CO_2$=406 ppmv, $O_3$=84 ppbv). The concentrations of VOCs for this sample are outliers relative the associated abundances of the trace gases. This anomaly is indicative of valve leakage during transit or ambient air entering the WAS canister after the flight. The observed CO to acetylene ratio (ppbv/ppbv), often used as a tracer for the age of an air mass, was much smaller in this sample (70 ppbv/ppbv) compared to other samples collected at a similar altitude (~400 ppbv/ppbv).

**Table S1. Summary statistics of the 1-second measured concentrations for O₃, NO₂, and CO, flight path descriptions, and weather conditions for each flight during ARIAs. Negative values of NO₂ indicate when the instrument was measuring around the detection limit.**

| Date (DOY) | Takeoff (LST) | Landing (LST) | Mean O₃ (Range), ppbv | Mean NO₂ (Range), ppbv | Mean NOy (Range), ppbv | Mean CO (Range), ppbv | Flight Description | Weather Conditions |
|---|---|---|---|---|---|---|---|---|
| May 8 (129) | 10:30 | 14:32 | 76.6 (62.7–83.9) | No data | 15.6 (9.0–29.4) | No data | Spirals over Julu (400-3500 m) at 10:58 LST, Quzhou (350-3500 m) at 12:00 LST, Xingtai (400-3000 m) at 12:23 LST, and Shijiazhuang (100-3500 m) at 14:05. | A high over the region, a weak low to the N, moving to the E. Strong winds in daytime (surface wind speed up to 10 m/s. Cold front passed 2 days prior. |
| May 15 (136) | 12:17 | 15:04 | 64.6 (54.4–85.8) | No data | No data | No data | Spirals over Julu (400-3500 m) at 12:43 LST and Quzhou at 13:41. | A high over the region, an occluded front to the over the Yellow Sea. Surface winds mostly the NE up to 10 m/s |
| May 16 (137) | 15:03 | 15:54 | 85.3 (70.5–96.0) | No data | 22.8 (6.1–29.6) | No data | Flight to the southeast to the W of Julu. Flight altitude about 400 m. | A high over the region, a cold front to the E of Korean Peninsula. Mon surface winds from the (< 8 m/s) with a shift morning from the SE ( m/s). |
| May 17 (138) | 8:21 | 11:13 | 80.1 (45.0–99.1) | 8.8 (−0.1–38.4) | 30.2 (0.2–89.7) | 590.5 (114.3–6053.6) | Low altitude transect to Julu, with spirals at Julu (650-2800 m) at 9:47 LST and Quzhou (400-3000 m) at 10:19 LST. | Surface high pressure conditions throughout region and low pressure the N and W. A 700 ridge is situated over Korea. Surface W wind the morning (< 5 m/s) a shift in late-morning f the SE (< 10 m/s). |
| May 19 (140) | 15:42 | 17:09 | 97.1 (75.4–130.2) | 1.4 (−0.1–6.8) | 11.3 (3.6–29.1) | 131.4 (90.8–540.0) | Spirals over the airport. | Weak surface high pres conditions over the reg and low pressure syste over S Mongolia and I Mongolia. Upper level hPa trough from the previous day moved Sea of Okhotsk. Surfa winds from the W in morning (< 5 m/s) wi shift in late-morning fr the SE (< 7 m/s). |
| May 21 (142) | 11:57 | 13:41 | 99.5 (67.1–145.6) | 1.9 (−0.1–16.4) | No data | 238.5 (80.5–564.5) | Flew to southeast at low altitude (1000 m) to a point (114.9 °E, 37.6 °N). Spirals over Quzhou (300-3000 m) at 12:40 LST and Xingtai (300-2400 m) at 13:34 LST. | Weak surface high pres conditions over the reg with a Siberian Anticyc to the N. Surface win from the W in the morn with a shift in late-mor from the E (< 6 m/s |
| May 28 (149) | 10:16 | 13:26 | 86.3 (63.5–100.3); 8.9 (72.9–112.3) | 3.2 (−0.1–27.0); 2.6 (0.01–10.4) | No data | 332.2 (97.1–1264.9); 215.2 (88.1–963.3) | Morning flight flew spirals over Xingtai (350-3000 m) at 11:02 LST and Julu (450-2500 m) at 12:29 LST. During the afternoon flight, spirals over Xingtai (350-3000 m) at 16:57 LST. | High pressure over th region and a stationa front ~1000 km to th near Shanghai. Surfac winds mostly from the (< 5 m/s). |
| | 16:29 | 18:24 | | | | | | |
| June 2 (154) | 13:47 | 14:53 | 94.9 (79.6–106.3) | 1.5 (−0.1–5.4) | 24.3 (14.9–70.7) | 256.7 (95.1–487.5) | Spirals over the airport. | High over the region low pressure over cen China and a stationary to the S near Shangha |

| | | | | | | | | |
|---|---|---|---|---|---|---|---|---|
| | | | | | | | Light surface winds (< 5 m/s) mostly from the W. | |
| June 6 (158) | 10:08 | 12:01 | 99.9 (67.5-134.7) | 0.7 (−0.1-4.9) | No data | 296.2 (105.1-573.2) | Low altitude (< 2000 m) spirals to the SE of Shijiazhuang. Spirals over Julu at 10:44 LST. | Several weak low pres... systems over the region... stationary front is over... East China Sea. Varia... winds less than 5 m/s |
| June 11 (163) | 11:02 | 13:45 | 76.7 (57.2-90.8) | 2.3 (−0.1-6.7) | 16.9 (11.1-23.8) | 187.4 (88.2-412.9) | Low altitude transect (2000 m) to NE Julu. Spirals over Xingtai (600-3000 m) at 11:54 LST and Shijiazhuang (600-3000 m) at 13:12 LST. | Low pressure over reg... with a Siberian anticyc... over Mongolia. A stationary front is locat... the S near Taiwan. Var... surface winds, with th... strongest winds (12 m... from the N in the morn... |

160

**Table S2. Summary statistics of alkanes, alkenes/alkynes, and aromatics quantified for all WAS canisters (pptv), as well as the method detection limit (MDL, in pptv), rate constants with OH (kOH), maximum incremental reactivity (MIR) value, and ratio to CO (pptv/ppbv) for compounds with R>0.50. Values less than 1 pptv are not shown.**

| | Mean (STD) | Min | 5th | 25th | 50th | 75th | 95th | Max | MDL* | kOH‡ | MIR† | |
|---|---|---|---|---|---|---|---|---|---|---|---|---|
| **Alkanes** | | | | | | | | | | | | |
| Ethane | 2648 (710) | 1804 | 1902 | 2033 | 2525 | 2998 | 4066 | 4154 | 50 | $6.90 \times 10^{-12} \times e^{-1000/T}$ | 0.28 | - |
| Propane | 1391 (231) | 978 | 1044 | 1196 | 1356 | 1509 | 1769 | 1887 | 21 | $7.60 \times 10^{-12} \times e^{-585/T}$ | 0.49 | - |
| n-Butane | 363 (278) | 83 | 92 | 207 | 259 | 480 | 1131 | 1210 | 30 | $9.80 \times 10^{-12} \times e^{-425/T}$ | 1.15 | - |
| 2,2-Dimethylbutane | 13 (14) | 2 | 3 | 5 | 9 | 17 | 42 | 64 | 7 | $3.22 \times 10^{-11} \times e^{-781/T}$ | 1.17 | - |
| 2,3-Dimethylbutane | 44 (93) | 2 | 2 | 5 | 11 | 27 | 293 | 400 | 5 | $1.24 \times 10^{-17} \times T^2 \times e^{-585/T}$ | 0.97 | - |
| i-Butane | 624 (997) | 56 | 70 | 109 | 246 | 673 | 3546 | 3963 | 29 | $1.16 \times 10^{-17} \times \times T^2 \times e^{225/T}$ | 1.23 | - |
| n-Pentane | 119 (113) | 19 | 26 | 54 | 71 | 155 | 400 | 479 | 5 | $2.44 \times 10^{-17} \times \times T^2 \times e^{183/T}$ | 1.31 | - |
| i-Pentane | 674 (1255) | 32 | 48 | 118 | 168 | 413 | 3785 | 5444 | 12 | $3.70 \times 10^{-12}$ | 1.45 | - |
| Cyclopentane | 34 (64) | 2 | 2 | 5 | 12 | 25 | 168 | 296 | 26 | $2.67 \times 10^{-11} \times e^{-590/T}$ | 2.39 | - |
| Methylcyclopentane | 26 (29) | 2 | 3 | 7 | 15 | 28 | 91 | 115 | 8 | $7.66 \times 10^{-12}$ | 2.19 | - |
| 2-Methylpentane | 111 (144) | 8 | 11 | 39 | 61 | 103 | 363 | 667 | 5 | $5.30 \times 10^{-12}$ | 1.5 | - |
| 3-Methylpentane | 53 (88) | 3 | 3 | 9 | 26 | 55 | 224 | 395 | 7 | $5.40 \times 10^{-12}$ | 1.8 | - |
| 2,3-Dimethylpentane | 27 (33) | 4 | 4 | 9 | 19 | 26 | 86 | 152 | 4 | $1.95 \times 10^{-11} \times e^{-330/T}$ | 1.34 | - |
| 2,4-Dimethylpentane | 31 (53) | 3 | 3 | 6 | 11 | 25 | 137 | 228 | 5 | $2.49 \times 10^{-11} \times e^{-443/T}$ | 1.55 | - |
| 2,2,4-Trimethylpentane | 433 (1117) | 9 | 11 | 28 | 57 | 236 | 2120 | 5422 | 3 | $2.09 \times 10^{-12} \times (\frac{T}{298})^{2.00} \times e^{140/T}$ | 1.26 | - |
| 2,3,4-Trimethylpentane | 232 (652) | 9 | 9 | 15 | 33 | 96 | 987 | 3253 | 8 | $9.85 \times 10^{-12} \times e^{-124/T}$ | 1.03 | - |
| n-Hexane | 123 (180) | 6 | 8 | 24 | 46 | 130 | 541 | 699 | 16 | $1.53 \times 10^{-17} \times T^2 \times e^{414/T}$ | 1.24 | - |
| Cyclohexane | 15 (13) | 1 | 2 | 5 | 9 | 27 | 44 | 44 | 16 | $2.88 \times 10^{-17} \times T^2 \times e^{-309/T}$ | 1.25 | - |
| Methylcyclohexane | 17 (23) | 3 | 5 | 6 | 10 | 14 | 54 | 114 | 8 | $1.18 \times 10^{-11}$ | 1.70 | - |
| 2-Methylhexane | 39 (72) | 6 | 8 | 13 | 18 | 27 | 147 | 362 | 8 | $6.86 \times 10^{-12}$ | 1.19 | - |
| 3-Methylhexane | 44 (88) | 7 | 7 | 12 | 16 | 33 | 178 | 438 | 6 | $7.15 \times 10^{-12}$ | 1.61 | - |

Formatted Table

| | | | | | | | | | | | | |
|---|---|---|---|---|---|---|---|---|---|---|---|---|
| n-Heptane | 41 (52) | 9 | 12 | 16 | 22 | 39 | 142 | 255 | 7 | $1.59 \times 10^{-17} \times T^2 \times e^{478/T}$ | 1.07 | - |
| 2-Methylheptane | 399 (1106) | 11 | 13 | 32 | 63 | 172 | 1718 | 5515 | 8 | $2.51 \times 10^{-17} \times T^2 \times e^{447/T}$ | 1.07 | - |
| 3-Methylheptane | 15 (13) | 7 | 7 | 8 | 11 | 14 | 56 | 59 | 9 | $2.51 \times 10^{-17} \times T^2 \times e^{447/T}$ | 1.24 | - |
| Octane | 26 (19) | 10 | 10 | 15 | 23 | 29 | 59 | 102 | 12 | $2.76 \times 10^{-17} \times T^2 \times e^{378/T}$ | 0.90 | - |
| n-Nonane | 22 (12) | 13 | 14 | 15 | 17 | 25 | 39 | 72 | 21 | $2.51 \times 10^{-17} \times T^2 \times e^{447/T}$ | 0.78 | - |
| n-Decane | 58 (57) | 14 | 14 | 24 | 38 | 76 | 155 | 288 | 10 | | 0.68 | |
| **Alkenes/Alkynes** | | | | | | | | | | | | |
| Acetylene | 803 (465) | 234 | 284 | 454 | 578 | 1175 | 1506 | 1934 | 48 | $1.69 \times 10^{-12} \times e^{-233/T}$ | 0.95 | 1.4 |
| Ethylene | 884 (923) | 185 | 191 | 281 | 405 | 1093 | 2941 | 3536 | 30 | $2.14 \times 10^{-12} \times e^{411/T}$ | 9.00 | 2.9 |
| Propylene | 168 (44) | 102 | 104 | 143 | 164 | 199 | 223 | 308 | 25 | | 11.66 | - |
| 1-Butene | 23 (10) | 10 | 11 | 17 | 19 | 25 | 43 | 46 | 30 | $6.60 \times 10^{-12} \times e^{465/T}$ | 9.73 | - |
| cis-2-Butene | 3 (6) | - | - | 1 | 1 | 2 | 7 | 31 | 23 | $1.10 \times 10^{-11} \times e^{487/T}$ | 14.24 | - |
| trans-2-Butene | 3 (3) | - | - | 1 | 2 | 4 | 9 | 16 | 31 | $1.01 \times 10^{-11} \times e^{550/T}$ | 15.16 | - |
| Isoprene | 35 (39) | 2 | 5 | 8 | 20 | 37 | 117 | 138 | 15 | $2.70 \times 10^{-11} \times e^{390/T}$ | 10.61 | - |
| 1-Pentene | 8 (3) | 4 | 4 | 6 | 7 | 9 | 14 | 18 | 9 | $5.86 \times 10^{-12} \times e^{500/T}$ | 7.21 | - |
| cis-2-Pentene | 2 (3) | - | - | 1 | 1 | 2 | 4 | 16 | 8 | $6.54 \times 10^{-11}$ | 10.38 | - |
| trans-2-Pentene | 2 (3) | - | - | 1 | 1 | 2 | 12 | 14 | 8 | $6.69 \times 10^{-11}$ | 10.56 | - |
| 1-Hexene | 6 (5) | 3 | 3 | 4 | 5 | 6 | 9 | 27 | 11 | $3.70 \times 10^{-11}$ | 5.49 | - |
| **Aromatics** | | | | | | | | | | | | |
| Benzene | 510 (521) | 63 | 96 | 188 | 330 | 570 | 1819 | 2183 | 7 | $2.30 \times 10^{-12} \times e^{-190/T}$ | 0.72 | 1.8 |
| Toluene | 757 (1188) | 31 | 43 | 159 | 300 | 627 | 4064 | 4402 | 5 | $1.80 \times 10^{-12} \times e^{340/T}$ | 4.00 | - |
| Styrene | 14 (12) | 4 | 4 | 6 | 8 | 18 | 43 | 45 | 13 | $5.80 \times 10^{-11}$ | 1.73 | - |
| m/p-Xylene | 108 (155) | 16 | 22 | 43 | 62 | 117 | 345 | 789 | 2 | $1.87 \times 10^{-11}$ | 7.80 | - |
| o-Xylene | 43 (51) | 8 | 11 | 17 | 28 | 44 | 119 | 263 | 3 | $1.36 \times 10^{-11}$ | 7.64 | - |
| Ethylbenzene | 73 (85) | 12 | 15 | 27 | 45 | 83 | 198 | 423 | 3 | $7.00 \times 10^{-12}$ | 3.04 | - |
| Isopropylbenzene | 15 (8) | 7 | 7 | 10 | 13 | 18 | 28 | 48 | 20 | $6.61 \times 10^{-12}$ | 2.52 | 0.02 |
| n-Propylbenzene | 15 (20) | 4 | 4 | 7 | 10 | 15 | 37 | 104 | 16 | $5.80 \times 10^{-12}$ | 2.03 | 0.06 |
| 2-Ethyltoluene | 14 (17) | 5 | 5 | 7 | 9 | 13 | 35 | 89 | 10 | $1.86 \times 10^{-11}$ | 5.59 | 0.05 |
| 3-Ethyltoluene | 19 (18) | 4 | 5 | 9 | 13 | 18 | 59 | 88 | 20 | $1.18 \times 10^{-11}$ | 7.39 | 0.05 |
| 4-Ethyltoluene | 19 (26) | 4 | 4 | 7 | 11 | 18 | 60 | 132 | 20 | $1.19 \times 10^{-11}$ | 4.44 | 0.07 |
| 1,3-Diethylbenzene | 20 (28) | 4 | 4 | 6 | 8 | 12 | 64 | 248 | 10 | $1.86 \times 10^{-11}$ | 7.10 | 0.15 |
| 1,4-Diethylbenzene | 27 (40) | 8 | 8 | 11 | 18 | 22 | 53 | 218 | 10 | $1.18 \times 10^{-11}$ | 4.43 | 0.12 |
| 1,2,3-Trimethylbenzene | 18 (24) | 7 | 7 | 9 | 13 | 16 | 43 | 130 | 2 | $3.27 \times 10^{-11}$ | 11.97 | 0.07 |
| 1,2,4-Trimethylbenzene | 30 (30) | 8 | 9 | 15 | 23 | 31 | 65 | 160 | 3 | $3.25 \times 10^{-11}$ | 8.87 | 0.08 |
| 1,3,5-Trimethylbenzene | 10 (11) | 3 | 3 | 4 | 6 | 8 | 28 | 54 | 4 | $5.67 \times 10^{-11}$ | 11.76 | - |

[*] TO-15 method, where the standard deviation of seven replicates near the detection limit are multiplied by 3.14 (Student's t value with 99% confidence).

[‡] Reaction rate coefficient with OH.

[†] Maximum Incremental Reactivity (MIR, units=g O₃/g VOC), from Carter, (2010).

190

**Figure S1. Left: Picture of the gas (alt-facing) and aerosol inlet (forward facing) on top of the Y-12 aircraft. Right: Picture**
195 **of the Cloud Water Inertial Probe (CWIP) on the Y-12 aircraft installed under the port wing.**

[Figure]

[Figure]

200

205

210

215

220

225

230

**Figure S2.** Scatter plot of 1-minute average $O_x$ ($O_3$+$NO_2$) as a function of $NO_z$ ($NO_y$-$NO_x$) less than 30 ppbv below 1500 m. The color shows the local hour of collection. The line is the linear regression with the slope (k) and Pearson R correlation coefficient.

235

[Figure]

**Figure S3. Vertical profiles (N=19) of 1-second O$_3$ concentrations (ppbv) from the Y-12 (circles) compared to concurrent average concentrations measured at the A$^2$BC site in Xingtai (diamonds). The average surface O$_3$ concentration was computed by averaging the 5-minute data interval starting 30 minutes before the spiral until 30 minutes after the spiral was completed.**

[Figure]

245

250

255

**Figure S4. Scatter plot of 1-second CO (ppbv) and CO₂ (ppmv) (left) and SO₂ (ppbv) and CO₂ (right) sampled during a plume over Julu on June 6.**

[Figure]

[Figure]

| | | |
|---|---|---|
| Page 2: [1] Deleted | Sarah Elizabeth Benish | 8/26/20 11:10:00 AM |
| Page 2: [1] Deleted | Sarah Elizabeth Benish | 8/26/20 11:10:00 AM |
| Page 2: [2] Deleted | Sarah Elizabeth Benish | 8/26/20 11:12:00 AM |
| Page 2: [2] Deleted | Sarah Elizabeth Benish | 8/26/20 11:12:00 AM |
| Page 2: [2] Deleted | Sarah Elizabeth Benish | 8/26/20 11:12:00 AM |
| Page 2: [3] Deleted | Sarah Elizabeth Benish | 8/26/20 11:12:00 AM |
| Page 2: [3] Deleted | Sarah Elizabeth Benish | 8/26/20 11:12:00 AM |
| Page 2: [4] Deleted | Sarah Elizabeth Benish | 8/26/20 11:13:00 AM |
| Page 2: [4] Deleted | Sarah Elizabeth Benish | 8/26/20 11:13:00 AM |
| Page 2: [5] Deleted | Sarah Elizabeth Benish | 8/26/20 11:13:00 AM |
| Page 2: [5] Deleted | Sarah Elizabeth Benish | 8/26/20 11:13:00 AM |
| Page 2: [6] Deleted | Sarah Elizabeth Benish | 8/26/20 11:14:00 AM |
| Page 2: [6] Deleted | Sarah Elizabeth Benish | 8/26/20 11:14:00 AM |
| Page 2: [7] Deleted | Sarah Elizabeth Benish | 8/26/20 11:14:00 AM |
| Page 2: [7] Deleted | Sarah Elizabeth Benish | 8/26/20 11:14:00 AM |
| Page 2: [8] Deleted | Sarah Elizabeth Benish | 8/26/20 11:15:00 AM |
| Page 2: [8] Deleted | Sarah Elizabeth Benish | 8/26/20 11:15:00 AM |
| Page 2: [9] Deleted | Sarah Elizabeth Benish | 8/26/20 11:15:00 AM |
| Page 2: [9] Deleted | Sarah Elizabeth Benish | 8/26/20 11:15:00 AM |
| Page 2: [9] Deleted | Sarah Elizabeth Benish | 8/26/20 11:15:00 AM |
| Page 2: [10] Deleted | Sarah Elizabeth Benish | 8/26/20 11:15:00 AM |
| Page 2: [10] Deleted | Sarah Elizabeth Benish | 8/26/20 11:15:00 AM |
| Page 2: [11] Deleted | Sarah Elizabeth Benish | 8/26/20 11:16:00 AM |
| Page 2: [11] Deleted | Sarah Elizabeth Benish | 8/26/20 11:16:00 AM |
| Page 2: [12] Deleted | Sarah Elizabeth Benish | 8/26/20 11:16:00 AM |
| Page 2: [12] Deleted | Sarah Elizabeth Benish | 8/26/20 11:16:00 AM |
| Page 2: [13] Deleted | Sarah Elizabeth Benish | 8/26/20 11:16:00 AM |
| Page 2: [13] Deleted | Sarah Elizabeth Benish | 8/26/20 11:16:00 AM |
| Page 2: [14] Deleted | Sarah Elizabeth Benish | 8/26/20 11:17:00 AM |
| Page 2: [14] Deleted | Sarah Elizabeth Benish | 8/26/20 11:17:00 AM |
| Page 2: [15] Deleted | Sarah Elizabeth Benish | 8/26/20 11:17:00 AM |
| Page 2: [15] Deleted | Sarah Elizabeth Benish | 8/26/20 11:17:00 AM |
| Page 2: [15] Deleted | Sarah Elizabeth Benish | 8/26/20 11:17:00 AM |
| Page 2: [16] Deleted | Sarah Elizabeth Benish | 8/26/20 11:18:00 AM |
| Page 2: [16] Deleted | Sarah Elizabeth Benish | 8/26/20 11:18:00 AM |
| Page 2: [17] Deleted | Sarah Elizabeth Benish | 8/26/20 11:18:00 AM |
| Page 2: [17] Deleted | Sarah Elizabeth Benish | 8/26/20 11:18:00 AM |
| Page 2: [18] Deleted | Sarah Elizabeth Benish | 8/26/20 11:19:00 AM |
| Page 2: [18] Deleted | Sarah Elizabeth Benish | 8/26/20 11:19:00 AM |
| Page 2: [18] Deleted | Sarah Elizabeth Benish | 8/26/20 11:19:00 AM |
| Page 2: [18] Deleted | Sarah Elizabeth Benish | 8/26/20 11:19:00 AM |
| Page 2: [19] Deleted | Sarah Elizabeth Benish | 8/26/20 11:18:00 AM |
| Page 2: [19] Deleted | Sarah Elizabeth Benish | 8/26/20 11:18:00 AM |
| Page 2: [20] Deleted | Sarah Elizabeth Benish | 8/26/20 11:19:00 AM |

| Page 2: [20] Deleted | Sarah Elizabeth Benish | 8/26/20 11:19:00 AM |
|---|---|---|
| Page 2: [21] Deleted | Sarah Elizabeth Benish | 8/26/20 11:19:00 AM |
| Page 2: [21] Deleted | Sarah Elizabeth Benish | 8/26/20 11:19:00 AM |
| Page 2: [22] Deleted | Sarah Elizabeth Benish | 8/26/20 11:19:00 AM |
| Page 2: [22] Deleted | Sarah Elizabeth Benish | 8/26/20 11:19:00 AM |
| Page 2: [23] Deleted | Sarah Elizabeth Benish | 8/26/20 11:20:00 AM |
| Page 2: [23] Deleted | Sarah Elizabeth Benish | 8/26/20 11:20:00 AM |
| Page 2: [24] Deleted | Sarah Elizabeth Benish | 8/26/20 11:21:00 AM |
| Page 2: [24] Deleted | Sarah Elizabeth Benish | 8/26/20 11:21:00 AM |
| Page 2: [25] Deleted | Sarah Elizabeth Benish | 8/26/20 11:21:00 AM |
| Page 2: [25] Deleted | Sarah Elizabeth Benish | 8/26/20 11:21:00 AM |
| Page 2: [26] Deleted | Sarah Elizabeth Benish | 8/26/20 11:22:00 AM |
| Page 2: [26] Deleted | Sarah Elizabeth Benish | 8/26/20 11:22:00 AM |
| Page 2: [27] Deleted | Sarah Elizabeth Benish | 8/26/20 11:22:00 AM |
| Page 2: [27] Deleted | Sarah Elizabeth Benish | 8/26/20 11:22:00 AM |